# Chain-of-Learngene: A Scalable Learngene-based Paradigm for Building and Initializing Variable-Sized Small Language Models

## Abstract

Large language models (LLMs) show strong performance across a wide range of tasks, yet their deployment remains costly in resource-constrained environments. A common alternative is to pre-train small language models (SLMs) from scratch, but this approach demands substantial computation and often suffers from limited model capacity. Knowledge distillation (KD) improves SLMs' performance by transferring knowledge from LLMs, but generating SLMs of varying sizes typically requires repeated teacher (LLMs) inference, which remains computationally expensive. To address these challenges, we propose **Chain-of-Learngene (CoL)**, a scalable framework for efficiently initializing multi-scale SLMs for diverse resource-constrained settings. **CoL** is inspired by the Learngene framework, which extracts expressive and tiny components (*learngene*) from a pre-trained ancestor model (AnsNet) to initialize descendant models (DesNets) of different sizes. Building on this idea, **CoL** constructs a sparse sequence of intermediate models, forming a *learngene chain*, through a few stepwise distillation steps from the AnsNet. Besides, a *bridge distillation* mechanism is introduced to support AnsNets with different architectures or vocabularies. Finally, **CoL** initializes variable-sized SLMs via parameter interpolation between adjacent models in the chain, thereby eliminating duplicate access to the LLMs. Experiments show that **CoL** significantly improves efficiency, scalability, and downstream performance. For instance, a 138M DesNet initialized by **CoL** without any recovery pre-training outperforms scratch-trained models on a 10B-token corpus.

## 1 Introduction

Large language models (LLMs) (Touvron et al., 2023; Team, 2024; Yang et al., 2025; Li et al., 2025) demonstrate strong generalization and have been widely adopted across vertical domains (Cui et al., 2023; Chen et al., 2023). However, their adaptability often requires intensive resource support, posing challenges for deployment in constrained settings such as mobile or in-vehicle systems (Jia et al., 2025). Moreover, the diversity of real-world applications leads to varying resource constraints, calling for a range of smaller language models (SLMs) with varying sizes for different scenarios.

To efficiently support deployment across diverse resource-constrained environments, the most ideal method is to construct a densely populated set of SLMs covering all possible sizes, enabling direct retrieval for target specific constraints. A straightforward approach to achieve this is to individually pre-train each SLMs from scratch (Singer et al., 2024; Devlin et al., 2019). However, as shown in Figure 1a, this requires training $N$ separate models for $N$ target sizes, which is prohibitively time-consuming and resource-consuming. Moreover, smaller models often struggle to achieve strong performance due to limited model capacity. Knowledge distillation (KD) (Yao et al., 2023; Lupart et al., 2025; Hu et al., 2025; Du et al., 2025) can improve the performance of the pre-trained SLMs by training SLMs to imitate the larger teacher model. However, this approach has two major limitations. Firstly, it requires repeated access to the teacher LLMs to generate supervision signals, which is computationally expensive and becomes increasingly costly as the number of SLMs grows, as shown in Figure 1b.

Secondly, when the student SLMs are significantly smaller than the teacher LLMs, the large capacity gap can hinder effective knowledge transfer, resulting in degraded student performance (Muralidharan et al., 2024; Zhang et al., 2023). Beyond deployment, Neural Architecture Search (NAS) Cai et al. (2018) pipelines also require evaluating large numbers of candidate models across different sizes. Current proxy-based Mellor et al. (2021); Li et al. (2023), early-stopping strategies Klein et al. (2017) or sampling sub-models from a supernet Shi et al. (2020) often fail to provide reliable estimates due to weak initialization. Thus, a method that can efficiently generate well-initialized models at many scales is highly desirable. These motivate a fundamental question: **how can we build a scalable and resource-efficient paradigm to rapidly construct smaller models across multiple sizes?**

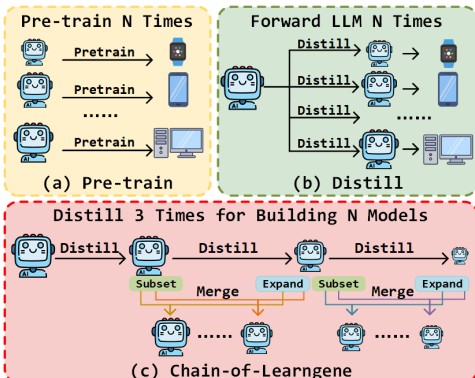

Figure 1: Comparison of pre-training, distillation, and **CoL** for building $N$ variable-sized models.

A recent paradigm known as *Learngene* (Wang et al., 2022a) addresses the above challenge of scalable model construction and initialization by extracting compact and informative parts, termed the "learngene", from a large pre-trained model (termed as the ancestor model, AnsNet) to initialize smaller models (termed as descendant models, DesNets), as illustrated in Figure 2a. This method saves training costs by forwarding the AnsNet **once** to extract the learngene and adopts it to initialize multiple DesNets for better performance. However, previous *Learngene* approaches rely on a single compact model. They extract critical information from AnsNet into one small module and then scale it to various model sizes through replication or interpolation. In the context of LLMs, this **one-size-fits-all** strategy presents two significant limitations. Firstly, the compact learngene often fails to retain the full semantic capacity of the large AnsNet, making it a suboptimal foundation for expansion. Secondly, directly scaling from this single learngene to multiple sizes results in unstable initialization and performance degradation due to the absence of representation from intermediate-sized parts.

To address these limitations, we propose a novel paradigm called **Chain-of-Learngene (CoL)**, which is efficient and scalable for building variable-sized SLMs. Firstly, **CoL** constructs a sparse sequence of intermediate models (termed as checkpoints) to form a *learngene chain* through the **stepwise distillation** from the AnsNet. Different the traditional KD, **stepwise distillation** distills the largest checkpoint from the AnsNet, and then each smaller checkpoint is distilled from its immediate predecessor in the chain, as shown in Figure 1c.

Moreover, to support heterogeneous AnsNets with different vocabularies or architectures (e.g., Qwen3-4B (Yang et al., 2025) versus GPT2-based chains), we introduce an bridging mechanism. Specifically, the AnsNet is first distilled into a structurally aligned proxy model (e.g., GPT2-XL) that matches the learngene chain architecture, serving as the starting point for stepwise distillation. This strategy narrows the capacity gap between the AnsNet and the smallest checkpoint, improving training stability and knowledge retention. Here, we theoretically analyze the benefits of stepwise distillation in improving convergence rates and reducing approximation errors, and further validate it by empirical results.

Fianlly, for building DesNets with variable sizes, we select two adjacent checkpoints in the chain and interpolate their parameters: the weights of the smaller checkpoint are expanded, those of the larger checkpoint are subseted. Then, the both weights are added with a scaling coefficient $\alpha$. This process enables DesNets to be constructed without repeatedly accessing the full AnsNet, significantly reducing computational overhead while maintaining high initialization quality.

Unlike conventional pre-training or KD methods that require pre-training or distilling $N$ separate SLMs for $N$ target sizes, **CoL** only distills $M = 3$ $(M \ll N)$ checkpoints from the AnsNet. These checkpoints form the *learngene chain* and provide efficient initialization of the remaining $(N - M)$ models. This method substantially reduces training costs from $\mathcal{O}(N)$ to $\mathcal{O}(1)$ while maintaining strong performance. Furthermore, different from the previous Learngene methods that rely on a single compact model, **CoL** leverages multiple intermediate-sized models to capture a broader range

of semantic information from the AnsNet. This multi-point representation enhances the quality of initialization for various DesNets. Extensive experiments demonstrate the remarkable effectiveness of **CoL** in improving training efficiency and performance. For instance, a 138M DesNet initialized via **CoL**, even without further training, outperforms scratch-trained models on **10B** training tokens. Besides, **CoL** dramatically accelerates convergence across multiple model scales and token budgets. For example, on the 500M tokens, **CoL** enables DesNet-138M to reach comparable performance to training from scratch, while achieving up to a **200×** speedup, significantly reducing training cost.

## 2    RELATED WORK

### 2.1    LLM KNOWLEDGE DISTILLATION

Knowledge distillation (Hinton et al., 2015) transfers knowledge from a large teacher model to a smaller student, enabling efficiency without sacrificing performance. In LLMs, methods are generally categorized as *black-box* or *white-box* distillation based on teacher accessibility. **Black-box distillation** targets closed-source models where internal states are unavailable. Student models are trained on outputs generated by the teacher, such as chain-of-thought (CoT) reasoning (Wang et al., 2022b; Hsieh et al., 2023; Wang et al., 2023a) or in-context learning (ICL) examples (Li et al., 2024; Duan et al., 2024). **White-box distillation** (Gu et al., 2023) uses open-source teachers. Beyond aligning output logits (Agarwal et al., 2024), students can learn from intermediate features (Liang et al., 2023), enabling fine-grained transfer.

### 2.2    LEARNGENE

Learngene-based methods (Wang et al., 2022a) extract compact, reusable subnetworks (*learngene*) from a large model (AnsNet) to initialize smaller descendant models (DesNets), followed by minimal fine-tuning. Vanilla Learngene (Van-LG) (Wang et al., 2022a) selects high-level layers based on gradient signals. Auto-Learngene (Wang et al., 2023b) uses pseudo DesNets and a meta-network to automatically select learngene layers. Learngene Pool (Shi et al., 2024) forms a layer pool by distilling multiple small models from the teacher. TLEG (Xia et al., 2024b) and PEG (Wang et al., 2024) expand selected layers linearly or probabilistically. SWS (Xia et al., 2024a) builds a multi-stage student model where each stage serves as a modular learngene.

While these learngene methods extract or expand compact subnetworks to initialize smaller models, they can not generate multiple high-quality SLMs across diverse target size like **CoL**. We also note that Chain-of-Model (CoM) (Wang et al.) uses a superficially similar chain concept. However, CoM enables a single backbone to support multi-scale inference and therefore differs fundamentally in purpose and design, which is different from **CoL**.

## 3    METHODOLOGY

To build variable-sized DesNets for downstream tasks with different resource constraints, we propose a novel method named **Chain-of-Learngene (CoL)**. The overall framework of **CoL** is illustrated in Figure 2. **CoL** consists of two main stages: constructing the learngene chain, and efficiently building DesNets with varying sizes from the **CoL**.

### 3.1    THE IMPLEMENTATION OF THE LEARNGENE CHAIN

The learngene chain is a chain-structured framework composed of multiple language models of different sizes, each referred to as a checkpoint. These checkpoints are arranged in descending order of model size, forming a progressively transitioning pathway of knowledge. In this work, we adopt {GPT2-L (762M), GPT2-M (345M), and GPT2-B (117M)} as the checkpoints of the GPT2-based learngene chain, {Pythia-1b, Pythia-410m, Pythia-160m, and Pythia-70m} Biderman et al. (2023) as the checkpoints of Pythia-based chain, and {Qwen3-1.7B, Qwen3-0.6B} Yang et al. (2025) as the checkpoints of Qwen3-based chain. For brevity, we mainly use GPT2-based learngene chain for description in the following sub-sections. The choice of these models is driven by two primary considerations. Firstly, they form a sequence of structurally consistent models with smoothly decreasing

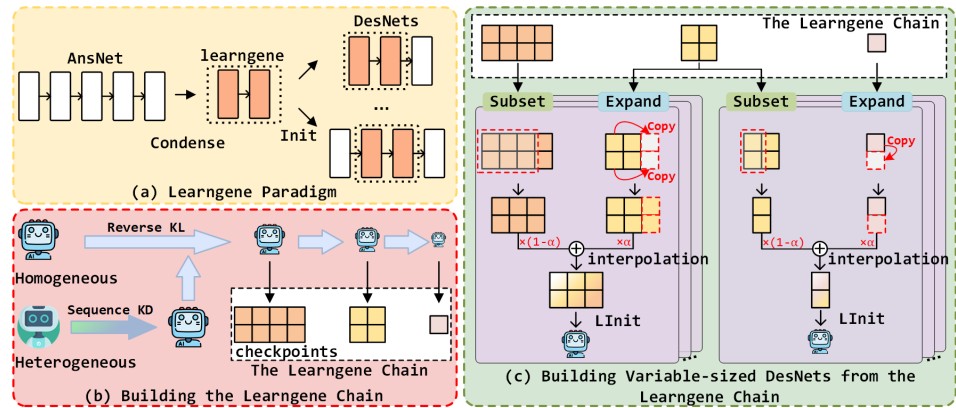

Figure 2: The overall structure of the Learngene paradigm and the proposed **CoL**.

sizes, which facilitates stable stepwise distillation. Subsequently, all models are publicly available, fully pre-trained, and widely adopted, thus avoiding the need for costly training from scratch.

## 3.2 BUILDING THE LEARNGENE CHAIN

Once the structure of the learngene chain is established, it is essential to propagate effective knowledge from the AnsNet to form a coherent chain. We propose two distillation strategies to accommodate both homogeneous and heterogeneous AnsNets, as illustrated in Figure 2b.

**Homogeneous Case.** When the AnsNet shares the same architecture as the checkpoints in the learngene chain (e.g., GPT2-XL, 1.5B (Radford et al., 2019)), we employ a sequential stepwise distillation strategy. Beginning with GPT2-XL, the knowledge is transferred successively to GPT2-L, GPT2-M, and finally GPT2-B. At each step, the preceding checkpoints acts as the teacher, while the current checkpoint serves as the student. The training is conducted on a general supervised fine-tuning (SFT) dataset $\mathcal{D}$, using the reverse Kullback-Leibler (KL) divergence (Gu et al., 2023) as the distillation objective:

$$\mathcal{L}_{\text{distill}} = \mathbb{E}_{x \sim \mathcal{D}} \left[ \sum_t P_{\text{student}}(x_t \mid x_{<t}) \log \frac{P_{\text{student}}(x_t \mid x_{<t})}{P_{\text{teacher}}(x_t \mid x_{<t})} \right] \tag{1}$$

Here, $x$ is an input sequence sampled from the training dataset $\mathcal{D}$. The term $x_{<t}$ is the context tokens preceding time step $t$, and $x_t$ denotes the target token at position $t$. $P_{\text{student}}(x_t \mid x_{<t})$ and $P_{\text{teacher}}(x_t \mid x_{<t})$ refer to the predicted probability distributions over the next token given the context produced by the student and teacher models. Reverse KL emphasizes alignment on high-confidence predictions, encouraging the student to replicate the teacher's capabilities more faithfully. This ensures the retention of strong knowledge representations within the chain.

**Heterogeneous Case.** When the architecture of AnsNet (e.g., Llama3-8B (Touvron et al., 2023) and Qwen3-4B (Yang et al., 2025)) differs from that of checkpoints in the learngene chain, we encounter two primary challenges arise: large model gap and incompatible vocabularies. These issues hinder direct knowledge transfer and degrade model alignment. To address them, we introduce the bridge distillation mechanism.

As shown in Figure 2b, we first distill the heterogeneous AnsNet into GPT2-XL using Sequence-level Knowledge Distillation (SeqKD). This process involves generating pseudo answers from the AnsNet and using the resulting question-answer pairs $(x, \hat{y})$ to fine-tune a compatible GPT2-XL model:

$$\mathcal{L}_{\text{seqkd}} = \mathbb{E}_{(x,\hat{y}) \sim \text{LM}_{\text{ansnet}}} \left[ -\log P_{\text{bridge}}(\hat{y} \mid x) \right] \tag{2}$$

Here, $x$ is the input prompt, and $\hat{y}$ is the corresponding response generated by the heterogeneous AnsNet. $P_{\text{bridge}}(\hat{y} \mid x)$ represents the probability assigned by the bridge model to the generated answer $\hat{y}$ given the input $x$. The expectation is taken over the distribution of pseudo data produced by the AnsNet. The resulting GPT2-XL acts as a bridge model, and we then apply the aforementioned stepwise distillation to complete the knowledge propagation within the learngene chain. This bridging strategy alleviates structural mismatches and mitigates the difficulty of direct distillation into

the target checkpoint from the Large AnsNet. Moreover, it improves training stability by avoiding abrupt capacity transitions, thereby enhancing the reliability and generalization of DesNets.

### 3.3 STEPWISE DISTILLATION

**Why Stepwise Distillation?** We adopt the stepwise distillation strategy for several key reasons. Firstly, it enhances training stability by ensuring that each distillation step occurs between models with relatively small architectural and capacity gaps. Figure 5 compares the training stability of our stepwise distillation with direct distillation. Our method yields smoother loss curves with lower variance, demonstrating its effectiveness in narrowing inter-model gaps and improving training stability. Secondly, this approach enables all checkpoints in the chain to learn shared general capabilities from the same supervised dataset, thereby promoting consistency among checkpoints. Lastly, stepwise distillation facilitates the progressive refinement of knowledge, allowing intermediate models to better capture and consolidate features before transferring them, which ultimately leads to improved knowledge retention and enhanced downstream performance.

**Theoretical Analysis of Stepwise Distillation.** We adopt the stepwise distillation strategy based on the assumption that directly distilling the large LLM into a smaller one suffers from poor convergence rates and large approximation gaps. By contrast, introducing intermediate models decomposes this large capacity gap into several smaller steps, thereby improving statistical rates and reduces approximation error. The following theorems prove our assumptions.

**Theorem 3.1** (Homogeneous stepwise distillation). *Consider a $k$-level distillation chain $\mathcal{H}_0 \to \mathcal{H}_1 \to \cdots \to \mathcal{H}_k$, where all $\mathcal{H}_i$ belong to the same hypothesis family. The generalization error of the final student satisfies*

$$R(\hat{f}_{\mathcal{H}_k}) - R(f^\star) \ \leq \ \sum_{i=1}^{k} O\Big(\frac{C_{\mathcal{H}_i}}{n^{\rho_{\mathcal{H}_i \leftarrow \mathcal{H}_{i-1}}}}\Big) + \sum_{i=1}^{k} \varepsilon_{\mathcal{H}_i \leftarrow \mathcal{H}_{i-1}}.$$

**Theorem 3.2** (Heterogeneous stepwise distillation). *Consider a $k$-level distillation chain $\mathcal{G} \to \mathcal{H}_1 \to \cdots \to \mathcal{H}_k \to \mathcal{S}$, where $\mathcal{G}$ and $\mathcal{S}$ may belong to different hypothesis families. The generalization error of the final student satisfies*

$$R(\hat{f}_{\mathcal{S}}) - R(f^\star) \ \leq \ \sum_{i=1}^{k} O\Big(\frac{C_{\mathcal{H}_i}}{n^{\rho_{\mathcal{H}_i \leftarrow \mathcal{H}_{i-1}}}}\Big) + O\Big(\frac{C_{\mathcal{S}}}{n^{\rho_{\mathcal{S} \leftarrow \mathcal{H}_k}}}\Big) + \sum_{i=1}^{k} \varepsilon_{\mathcal{H}_i \leftarrow \mathcal{H}_{i-1}} + \varepsilon_{\mathcal{S} \leftarrow \mathcal{H}_k}.$$

Specifically, as shown in Theorem 3.1, the statistical error and approximation error accumulate additively across $k$ homogeneous levels. Theorem 3.2 further extends this to the heterogeneous case, where an additional transfer term arises between the last assistant $\mathcal{H}_k$ and the student $\mathcal{S}$. In both cases, the bound is asymptotically tighter than that of direct distillation, supporting the intuition that breaking down the capacity gap stabilizes training. The detailed proofs and notations are provided in Appendix B.2 and Appendix B.3, respectively. Empirical results in Section 4.2 further confirm the theoretical analysis: stepwise distillation achieves higher accuracy and more stable training.

### 3.4 CONSTRUCTING VARIABLE-SIZED MODELS FROM THE LEARNGENE CHAIN

**CoL** aims to enable the rapid construction and initialization of target DesNets with variable sizes. To achieve this, we propose a parameter interpolation-based initialization method from the learngene chain (LInit), as illustrated in Figure 2c. Given a target-size DesNet, we first determine its relative position within the learngene chain and identify its two adjacent checkpoints. Parameters from the smaller node are expanded, while those from the larger checkpoint are compressed. These transformed parameters are then scaled by coefficients $\alpha$ and $(1 - \alpha)$, respectively, and linearly combined to initialize the DesNet.

For example, to initialize DesNet-138M (14L, 12H, 768D), we interpolate between its neighboring checkpoints: GPT2-B (12L, 12H, 768D) and GPT2-M (24L, 16H, 1024D). Here, **L**, **H**, and **D** represent the number of layers, attention heads, and hidden sizes. Specifically, we replicate the last 2 layers of GPT2-B and extract the first 14 layers of GPT2-M, adjusting attention heads and hidden sizes as needed. The resulting components are merged using weights $\alpha = 0.9$ and $0.1$, based on the relative distance of DesNet to its neighbors, assigning higher weight to the closer checkpoint. This

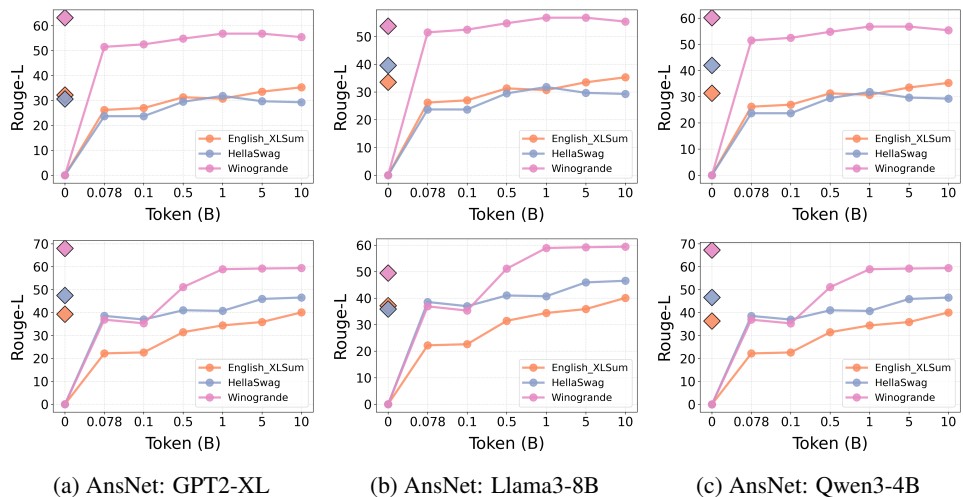

(a) AnsNet: GPT2-XL     (b) AnsNet: Llama3-8B     (c) AnsNet: Qwen3-4B

Figure 3: The number of tokens saved by LInit (Diamond) compared to training from scratch across three tasks. The DesNets in the first row are 138M, while the DesNets in the second row are 380M.

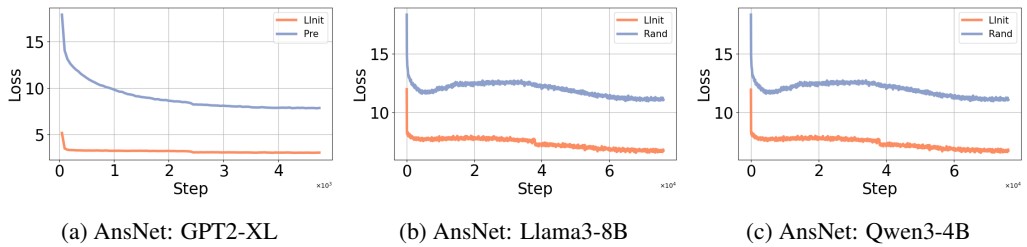

(a) AnsNet: GPT2-XL     (b) AnsNet: Llama3-8B     (c) AnsNet: Qwen3-4B

Figure 4: Convergence speed of DesNet-138M initialized by **CoL** (LInit) and from scratch (Rand) across 78M pre-training Tokens.

approach ensures that DesNets inherit the most relevant knowledge from their nearest checkpoints, facilitating efficient and effective initialization. We verify this by providing the performance of DesNets with the change of $\alpha$ in the Section 4.2.

# 4 EXPERIMENTS

To evaluate the proposed initialization method from the **CoL**, we experiment with three AnsNets: GPT2-XL (Radford et al., 2019), Llama3-8B (Touvron et al., 2023), Qwen3-4B (Yang et al., 2025), Pythia-1.4B Biderman et al. (2023), and generate DesNets at five scales (138M–537M). We benchmark on MMLU (Hendrycks et al., 2020), English_XSum (Narayan et al., 2018), HellaSwag (Zellers et al., 2019), WinoGrande (ai2, 2019), BoolQ (Clark et al., 2019), and Dolly (Conover et al., 2023). Besides, we also provide Pythia-based learngene chain to further verify the effectiveness of the proposed **CoL**. All checkpoints in **CoL** are distilled by 5000 steps, and the initialized DesNets are fine-tuned on the downsteam tasks for 10 epochs. The detailed prompts of the downstream tasks are shown in Appendix C. The other detailed experimental implementation is provided in Appendix C, the dataset information is summarized in Appendix D, and the architectures of the DesNets are described in Appendix E.

## 4.1 MAIN RESULTS AND ANALYSIS

In this subsection, we evaluate **CoL** initialization (LInit) from three perspectives: (1) pre-training corpora savings, (2) accelerated convergence during pre-training, (3) performance comparison with Random initialization and the previous Learngene method under equal data budgets, and (4) Additional results on Pythia-based and Qwen3-based learngene chain. Note that unless specified, all reported CoL results refer to the GPT-2–based chain for convenience.

Table 1: Comparison of **LInit** (learngene initialization) vs. **Rand** (random initialization) across various DesNet under varying token budgets. Tasks ①–⑤ correspond to MMLU, English_XLsum, HellaSwag, WinoGrande, and BoolQ.

| T | AnsNet: GPT2-XL | | | | AnsNet: Llama3-8B | | | | AnsNet: Qwen3-4B | | | |
|---|---|---|---|---|---|---|---|---|---|---|---|---|
| | 78M Token | | 100M Token | | 78M Token | | 100M Token | | 78M Token | | 100M Token | |
| | **LInit** | Rand | **LInit** | Rand | **LInit** | Rand | **LInit** | Rand | **LInit** | Rand | **LInit** | Rand |
| DesNet-138M | | | | | | | | | | | | |
| ① | 31.00 | 29.00 | 30.90 | 29.10 | 24.85 | 29.00 | 24.77 | 29.10 | 26.00 | 29.00 | 27.90 | 29.10 |
| ② | 30.68 | 26.18 | 31.54 | 26.95 | 34.22 | 26.18 | 34.20 | 26.95 | 29.83 | 26.18 | 29.83 | 26.95 |
| ③ | 31.15 | 23.68 | 30.94 | 23.68 | 35.51 | 23.68 | 35.17 | 23.68 | 42.65 | 23.68 | 41.50 | 23.68 |
| ④ | 62.00 | 51.50 | 61.90 | 52.50 | 44.41 | 51.50 | 42.90 | 52.50 | 57.60 | 51.50 | 55.20 | 52.50 |
| ⑤ | 76.80 | 68.00 | 76.60 | 67.80 | 64.97 | 68.00 | 64.33 | 67.80 | 68.70 | 68.00 | 66.00 | 67.80 |
| Avg | **46.33** | 39.67 | **46.38** | 40.01 | **40.79** | 39.67 | **40.27** | 40.01 | **44.96** | 39.67 | **44.09** | 40.01 |
| DesNet-220M | | | | | | | | | | | | |
| ① | 28.50 | 28.40 | 28.20 | 29.30 | 24.78 | 28.40 | 25.09 | 29.30 | 28.40 | 28.40 | 28.00 | 29.30 |
| ② | 26.68 | 28.48 | 28.27 | 26.87 | 32.77 | 28.48 | 34.79 | 26.87 | 24.10 | 28.48 | 30.45 | 26.87 |
| ③ | 26.84 | 23.87 | 28.76 | 23.67 | 35.37 | 23.87 | 35.65 | 23.67 | 41.41 | 23.87 | 43.49 | 23.67 |
| ④ | 58.00 | 49.70 | 59.20 | 52.80 | 46.20 | 49.70 | 43.22 | 52.80 | 55.20 | 49.70 | 56.10 | 52.80 |
| ⑤ | 69.50 | 66.50 | 70.70 | 69.10 | 63.76 | 66.50 | 64.90 | 69.10 | 68.80 | 66.50 | 67.40 | 69.10 |
| Avg | 41.90 | 39.39 | **43.03** | 40.35 | **40.58** | 39.39 | **40.73** | 40.35 | **43.58** | 39.39 | **45.09** | 40.35 |
| DesNet-277M | | | | | | | | | | | | |
| ① | 28.20 | 26.40 | 31.30 | 27.90 | 25.86 | 26.40 | 24.88 | 27.90 | 25.31 | 26.40 | 28.00 | 27.90 |
| ② | 32.36 | 27.30 | 35.31 | 27.68 | 33.98 | 27.30 | 32.86 | 27.68 | 33.72 | 27.30 | 30.45 | 27.68 |
| ③ | 29.57 | 23.77 | 29.57 | 24.39 | 35.65 | 23.77 | 35.43 | 24.39 | 33.87 | 23.77 | 42.47 | 24.39 |
| ④ | 61.49 | 50.20 | 58.47 | 47.50 | 43.22 | 50.20 | 44.58 | 47.50 | 40.83 | 50.20 | 57.60 | 47.50 |
| ⑤ | 71.80 | 67.50 | 73.00 | 66.70 | 65.86 | 67.50 | 64.97 | 66.70 | 68.50 | 67.50 | 68.70 | 66.70 |
| Avg | **44.68** | 39.03 | **45.53** | 38.83 | **40.92** | 39.03 | **40.54** | 38.83 | **40.44** | 39.03 | **45.44** | 38.83 |
| DesNet-380M | | | | | | | | | | | | |
| ① | 30.90 | 26.70 | 28.80 | 28.50 | 24.72 | 26.70 | 24.45 | 28.50 | 24.20 | 26.70 | 28.30 | 28.50 |
| ② | 34.14 | 22.20 | 35.53 | 22.62 | 34.51 | 22.20 | 33.42 | 22.62 | 36.31 | 22.20 | 37.28 | 22.62 |
| ③ | 46.04 | 38.54 | 43.23 | 36.96 | 36.46 | 38.54 | 34.42 | 36.96 | 44.09 | 38.54 | 42.40 | 36.96 |
| ④ | 68.60 | 36.90 | 66.20 | 35.30 | 40.58 | 36.90 | 41.15 | 35.30 | 59.80 | 36.90 | 59.60 | 35.30 |
| ⑤ | 74.90 | 67.90 | 75.00 | 65.10 | 63.65 | 67.90 | 64.79 | 65.10 | 70.20 | 67.90 | 67.40 | 65.10 |
| Avg | **50.92** | 38.45 | **49.75** | 37.70 | **39.99** | 38.45 | **39.65** | 37.70 | **46.92** | 38.45 | **47.00** | 37.70 |
| DesNet-537M | | | | | | | | | | | | |
| ① | 28.10 | 26.50 | 26.30 | 25.60 | 24.75 | 26.50 | 25.52 | 25.60 | 25.30 | 26.50 | 28.40 | 25.60 |
| ② | 30.53 | 19.14 | 34.74 | 15.15 | 34.95 | 19.14 | 34.42 | 15.15 | 35.10 | 19.14 | 35.03 | 15.15 |
| ③ | 40.75 | 37.54 | 42.02 | 36.59 | 34.60 | 37.54 | 33.68 | 36.59 | 39.30 | 37.54 | 44.10 | 36.59 |
| ④ | 58.90 | 33.50 | 60.20 | 34.00 | 43.08 | 33.50 | 38.20 | 34.00 | 56.90 | 33.50 | 56.80 | 34.00 |
| ⑤ | 69.50 | 64.70 | 71.90 | 62.80 | 62.73 | 64.70 | 64.54 | 62.80 | 68.40 | 64.70 | 73.50 | 62.80 |
| Avg | **45.56** | 36.28 | **47.03** | 34.83 | **40.02** | 36.28 | **39.27** | 34.83 | **45.00** | 36.28 | **47.57** | 34.83 |

**Saving Training Corpora**. To assess the effectiveness of the initialization method (LInit) from **CoL** in reducing training data requirements while preserving strong performance, we compare LInit **without recovery training** against conventional pre-training on corpora of different sizes. The OpenWebText-100K and OpenWebText datasets (Gokaslan et al., 2019) serve as the pre-training corpora. Here, we evaluate performance on three downstream tasks using DesNet-138M and 380M.

As shown in Figure 3, DesNets initialized with LInit, without consuming any recovery tokens, consistently outperform their pre-trained counterparts even after billions of tokens. Notably, on the HellaSwag task, DesNet-138M initialized by LInit surpasses models pre-trained with up to 10B tokens across all AnsNets, thereby saving 10B tokens (blue lines in the first row of Figure 3).

Similarly, on the English_XLSum task, DesNet-380M initialized with LInit achieves comparable performance while saving 5B pre-training tokens across all AnsNets relative to training from scratch (orange lines in the second row of Figure 3). These findings demonstrate that LInit offers a strong initialization strategy while substantially reducing the required pre-training corpus. Detailed numerical values from Figure 3 are provided in Appendix F.

Table 2: Performance of DesNet-113.63M initialized by **CoL** (Pythia-based) and random initialization.

|  | BoolQ | Arc-E | WinoG | Avg |
|---|---|---|---|---|
| Rand | 66.36 | 23.68 | 47.72 | 45.92 |
| CoL | 65.82 | 28.07 | 48.84 | **47.58** |

Table 3: The performance of intermediate checkpoint in **CoL** (GPT2-Based) and direct distillation.

|  | AnsNet | L | M | B |
|---|---|---|---|---|
| Direct | 27.94 | - | - | 23.30 |
| CoL | 27.94 | 27.77 | 27.63 | **26.02** |

Table 4: Performance comparison on three datasets and three AnsNets across different DesNet sizes (138M, 220M, 277M). "single" denotes the baseline that expands the model solely.

| Method | DesNet-138M | | | | DesNet-220M | | | | DesNet-277M | | | |
|---|---|---|---|---|---|---|---|---|---|---|---|---|
|  | Dolly | MMLU | BoolQ | Avg | Dolly | MMLU | BoolQ | Avg | Dolly | MMLU | BoolQ | Avg |
| AnsNet: GPT2-XL | | | | | | | | | | | | |
| single | 22.28 | 24.67 | 67.64 | 38.20 | 12.33 | 25.26 | 65.54 | 34.38 | 11.98 | 25.45 | 66.71 | 34.71 |
| **CoL** | 22.10 | 29.90 | 74.10 | **42.03** | 15.89 | 25.90 | 66.70 | **36.16** | 14.72 | 26.30 | 68.40 | **36.47** |
| AnsNet: Llama3-8B | | | | | | | | | | | | |
| single | 23.83 | 25.28 | 70.13 | **39.75** | 13.03 | 25.51 | 65.50 | 34.68 | 13.26 | 24.88 | 67.39 | 35.17 |
| **CoL** | 23.64 | 25.35 | 67.14 | 38.71 | 16.43 | 25.53 | 67.32 | **36.43** | 15.83 | 24.58 | 66.64 | **35.69** |
| AnsNet: Qwen3-4B | | | | | | | | | | | | |
| single | 23.88 | 26.01 | 66.68 | 38.86 | 13.41 | 25.33 | 66.54 | 35.09 | 12.69 | 24.50 | 66.25 | 34.48 |
| **CoL** | 25.25 | 28.90 | 74.40 | **42.85** | 16.02 | 25.75 | 64.60 | **35.46** | 16.08 | 28.90 | 67.40 | **37.46** |

**Fast Convergence Speed**. We compare the convergence speed of LInit with pre-training from scratch (i.e., random initialization, Rand) using the same 78M pre-training tokens for DesNet-138M. As shown in Figure 4a, when the AnsNet is GPT2-XL, the loss at the first step of LInit is already lower than the loss at the final step of Rand on 78M tokens, highlighting the strong initialization for accelerating the converging speed provided by LInit. Similar results are observed when the AnsNet is Llama3-8B and Qwen3-4B, as shown in Figure 4b,c. Moreover, LInit consistently requires far fewer steps than Rand to reach the same loss value. We also show more comparison of convergence speed on various pre-training token budgets in Appendix G, and these results further verify the fast convergence speed of the LInit.

**Better Performance under Equal Token Budgets**. To evaluate the performance of LInit, we compare models initialized with LInit against those trained from scratch (Rand) under identical token budgets (78M and 100M). All models are subsequently fine-tuned on downstream tasks for final evaluation. The results are summarized in Table 1, which reports performance across five representative benchmarks: MMLU, English_XLSum, HellaSwag, WinoGrande, and BoolQ, as well as their average scores.

The average results clearly demonstrate that LInit provides a substantially stronger initialization. As shown in Table 1, LInit consistently outperforms the Rand baseline across all tasks, model sizes, and AnsNets. For example, using Qwen-4B as the AnsNet, DesNet-380M initialized with LInit achieves 37.28% on English_XLSum (②) with only 100M recovery training tokens, compared to 22.62% from scratch—a gain of nearly 15%. This improvement indicates that LInit effectively inherits valuable knowledge from the AnsNet, providing a strong initialization for DesNet and outperforming Rand even when only a small number of pre-training tokens are available. Moreover, the consistently high average performance across all DesNet scales and AnsNets highlights the versatility and robustness of LInit under varying resource constraints.

We also analyze the impact of the performance gap between LInit and Rand as the number of pre-training tokens increases. The trends are visualized in Appendix H. We observe that although Rand improves substantially with sufficient pre-training (up to 10B tokens), LInit still maintains superior performance, further confirming its effectiveness.

Furthermore, we compare LInit with **previous Learngene** methods, including Auto-Learngene (Wang et al., 2023b) and Vanilla-Learngene (Wang et al., 2022a) in Appendix I. The results show that LInit from **CoL** consistently outperforms both methods across multiple tasks and DesNet scales under the same recovery token budgets, further demonstrating the effectiveness of LInit. Furthermore,

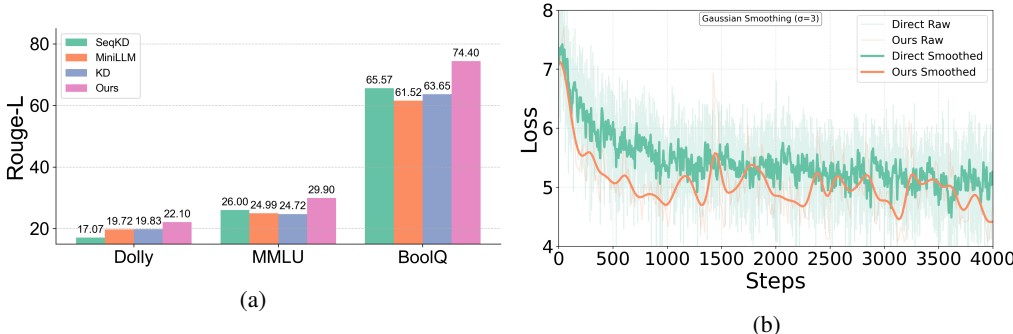

Figure 5: (a) Performance of DesNet-138M trained with LInit vs. KD baselines (teacher: GPT2-XL) on three tasks. (b) Training loss comparison between direct distillation and the stepwise distillation.

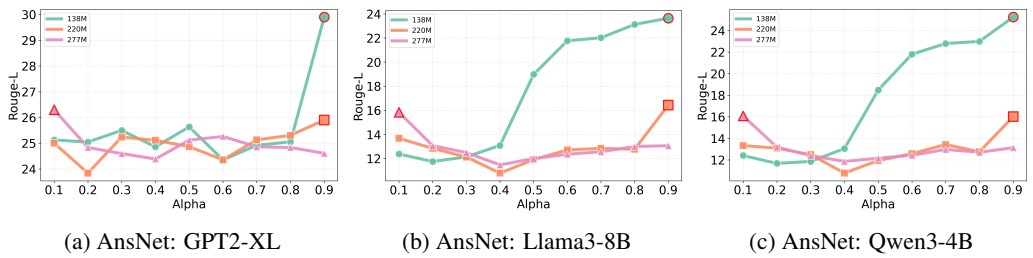

(a) AnsNet: GPT2-XL  (b) AnsNet: Llama3-8B  (c) AnsNet: Qwen3-4B

Figure 6: Performance comparison of various initialization coefficients ($\alpha$) for initialing DesNets.

since other Learngene-based methods developed for ViT architectures could not be successfully transferred to the LLM setting, our comparisons focus on Auto-Learngene and Vanilla-Learngene. The detailed explanation is provided in Appendix I.

Finally, we show the results of DesNet-113.63M initialized by Pythia-based **CoL**. As shown in Table 2, we find that DesNet-113.63M initialized by **CoL** still provides strong initialization compared to the random initialization. Qwen3-based **CoL** results (Appendix J) follow the same pattern observed with Pythia-based CoL. These verify the effectiveness of the **CoL** for different architectures.

## 4.2 ABLATION STUDIES

**CoL vs. Knowledge Distillation**. To further validate the effectiveness of our approach against standard knowledge distillation methods, we present a comparison in Figure 5a. Specifically, we evaluate DesNet-138M trained with three distillation methods: sequence-level KD (SeqKD), MiniLLM (Gu et al., 2023), and vanilla KD, all using the same teacher model (GPT2-XL). Across three benchmarks (Dolly, MMLU, and BoolQ), LInit consistently surpasses all distillation baselines by a substantial margin. These results highlight the superiority of LInit in transferring knowledge from larger teacher models, thereby improving downstream performance.

Furthermore, we compare **CoL** with direct distillation by evaluating intermediate checkpoints (Table 3). **CoL** maintains strong performance across all checkpoints, whereas direct distillation exhibits a sharp performance drop when distilling into GPT2-B. This indicates that stepwise distillation better preserves knowledge by narrowing the capacity gap, making **CoL** a more effective alternative to direct distillation.

Finally, We compare the training stability of stepwise distillation in the **CoL** with that of traditional knowledge distillation. For this analysis, we use GPT2-B as the student model and adopt two teacher models: GPT2-XL for the direct distillation baseline (Direct) and GPT2-M as the intermediate teacher in the **CoL**. The raw and smoothed training losses during distillation are shown in Figure 5b. We observe that **CoL** (orange line) produces a noticeably more stable training curve than the Direct baseline (green line) when the capacity gap between teacher and student is large. While traditional distillation often leads to instability, **CoL** transfers knowledge progressively through intermediate models of decreasing complexity, effectively bridging the semantic gap and enabling more stable and efficient learning.

Table 5: Performance comparison of DesNet-138M initialized by Single (+100M tokens) and GPT2-Based CoL (0 tokens).

|  | BoolQ | Arc-E | WinoG | Avg |
|---|---|---|---|---|
| Single | 66.11 | 26.04 | 58.06 | 50.07 |
| CoL | 74.40 | 25.70 | 63.20 | **54.43** |

Table 6: Downstream performance of Pythia-based CoL under different number of chain lengths.

| Length | BoolQ | Arc-E | WinoG | Avg |
|---|---|---|---|---|
| 2 | 65.60 | 24.21 | 48.92 | 46.24 |
| 4 | 65.82 | 28.07 | 48.84 | **47.58** |

**The Choice of Initialization Coefficient** $\alpha$. We examine the role of $\alpha$ in **CoL**, which interpolates parameters between smaller ($\alpha$) and larger ($1 - \alpha$) checkpoints in the learngene chain. As shown in Figure 6, we evaluate DesNet-138M, 220M, and 277M with $\alpha \in \{0.1, \ldots, 0.9\}$ on Dolly.

The results indicate that DesNets closer in scale to the smaller checkpoint benefit from larger $\alpha$ values (green/orange lines), whereas those closer to the larger checkpoint favor smaller $\alpha$ values (pink lines). This suggests that for DesNets architecturally closer to a particular checkpoint in the learngene chain, assigning greater weight to parameters derived from that checkpoint, either as a subset or an extension, enhances compatibility. Such a weighting strategy provides a stronger initialization, which in turn improves downstream task performance.

**CoL vs. Single Expansion**. To highlight the advantage of the interpolation strategy in **CoL**, we compare it with a *single-expansion* baseline that initializes DesNets only from the smallest checkpoint in the chain (GPT2-B). In contrast, **CoL** incorporates intermediate checkpoints (e.g., GPT2-M, GPT2-L), thereby providing richer initialization information.

The results show that although *Single* performs comparably to **CoL** on DesNet-138M with GPT2-XL as the AnsNet, its performance declines rapidly as the target DesNet size increases. Conversely, **CoL** consistently achieves superior results on larger models and datasets, yielding gains of +3.56% and +2.74% on Dolly for DesNet-220M and DesNet-277M under GPT2-XL, with similar improvements observed under Llama3-8B and Qwen3-4B.

We further compare **CoL** against a stronger Single baseline that receives an additional 100M recovery tokens before downstream fine-tuning. As shown in Table 5, even with substantially more training data, the Single model still performs worse than CoL, which requires **no** extra tokens (54.43 vs. 50.07). This result demonstrates that the initialization constructed from two adjacent checkpoints in **CoL** already contains sufficiently rich and well-aligned semantic information, enabling the descendant model to generalize more effectively than data-augmented Single. These findings suggest that relying solely on one checkpoint from the learngene chain constrains scalability, whereas **CoL** enables more effective transfer by leveraging multiple intermediate checkpoints.

**Various number of checkpoints**. We further study how the chain length affects downstream performance by fixing the same DesNet sizes to 113.63M and varying only the number of intermediate checkpoints. As shown in Table 6, a 4-checkpoint chain with smaller capacity gaps ($\approx$30–40%) outperforms a 2-checkpoint chain with larger gaps ($\approx$70–80%) on all benchmarks in average (47.58 vs. 46.24). This suggests that denser chains with smoother capacity transitions provide more effective initialization for CoL.

## 5 CONCLUSION

We introduce Chain-of-Learngene (**CoL**), a novel framework for the scalable and resource-efficient initialization of small language models (SLMs) across various sizes for resource-constraint environments. By progressively distilling knowledge through a structured chain of intermediate models, **CoL** eliminates the need for repeated access to the original large language models (LLMs), thereby significantly reducing the total training cost. To ensure architectural compatibility, a bridge distillation is introduced into the stepwise distillation, supporting the heterogeneous AnsNet for constructing the learngene chain. Furthermore, an interpolation-based technique is employed to initialize the target DesNets, achieving strong performance with minimal pretraining cost. Extensive experiments show that **CoL** is both effective and flexible, providing a practical solution for low-cost adaptation.

## ETHICS STATEMENT

This work complies with the ICLR Code of Ethics (https://iclr.cc/public/CodeOfEthics). It does not involve human subjects, personal data, or other sensitive information. All datasets used in the experiments are publicly available, and we adhered to standard practices for data usage and reporting. The methods and findings are intended solely for academic research and are not designed for harmful applications. We ensured that the experimental protocols, theoretical derivations, and released source code conform to the principles of fairness, transparency, and reproducibility. No conflicts of interest or ethical concerns are associated with this work.

## REPRODUCIBILITY STATEMENT

To ensure the reproducibility of this work, we provide the complete **source code as a zip archive** in the supplementary material. Appendix B contains a detailed derivation of the stepwise distillation theorem presented in the main paper, facilitating verification of the theoretical results. Appendix C specifies the experimental settings, including training details, evaluation metrics, and the hardware environment. In addition, Appendix D describes the downstream task datasets used for evaluation, enabling researchers to replicate the experimental results.

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

# A    APPENDIX

We organize the appendix as follows.

- In Section B, we list the theoretical analysis for the stepwise distillation under the homogeneous and heterogeneous AnsNets.
- In Section C, we illustrate the experimental details.
- In Section D, we describe the detailed information of datasets employed in the main paper.
- In Section E, we denote the detailed architectures of the DesNets with five sizes.
- In Section F, we show the detailed values in Figure 3 in the main paper.
- In Section G, we present an additional comparison of the convergence speed between DesNets initialized with LInit and those trained from scratch (Pre) under various token budgets.
- In Section H, we show the performance gap between the LInit and random initialization (Rand) with the increase of the pre-training tokens on the Dolly task. Here, all DesNet are built based on the AnsNet: GPT2-XL.
- In Section I, we compare the performance between the proposed initialization way from the **CoL** and **previous Learngene** method.
- In Section J, we demonstrate the results of Qwen3-Based **CoL** method.
- In section K, we discuss the similarities and differences with other scalable models.
- In section L, we show the Performance of CoL at Very Small and Very Large SLM Scales.
- In section M, we compare the performance of DesNets built from the AnsNet GPT2-XL and Qwen3-4B on the generation tasks.
- In Section N, we describe the precise role of the LLMs in this paper.
- In Section O, we show the limitation and future work of the proposed initialization way from the **CoL**.

# B    THEORETICAL ANALYSIS

We provide a unified theoretical analysis for $k$-level stepwise distillation as employed in CHAIN-OF-LEARNGENE. This section first introduces notation and the general single-step learning bound, then analyzes the homogeneous (same-architecture) case, and finally extends the results to the heterogeneous (cross-architecture) case.

## B.1    UNIFIED NOTATION AND PRELIMINARIES

Let the ground-truth target function be $f^\star$. We denote hypothesis classes in a distillation chain as

$$\mathcal{G} \;\to\; \mathcal{H}_1 \;\to\; \mathcal{H}_2 \;\to\; \cdots \;\to\; \mathcal{H}_k \;\to\; \mathcal{S},$$

where $\mathcal{G}$ is the ancestor teacher (possibly heterogeneous), $\mathcal{H}_i$ are intermediate assistants, and $\mathcal{S}$ is the final student.

For any hypothesis class $\mathcal{X}$ we denote:

- $C_\mathcal{X}$: a capacity measure;
- $\rho_{\mathcal{X}\leftarrow\mathcal{Y}}$: the statistical convergence rate when $\mathcal{X}$ learns from $\mathcal{Y}$;
- $\varepsilon_{\mathcal{X}\leftarrow\mathcal{Y}}$: the approximation error when $\mathcal{X}$ mimics $\mathcal{Y}$.

**General single-step bound.** For any pair $(\mathcal{X}, \mathcal{Y})$,

$$R(\hat{f}_\mathcal{X}) - R(f_\mathcal{Y}) \;\leq\; O\!\left(\frac{C_\mathcal{X}}{n^{\rho_{\mathcal{X}\leftarrow\mathcal{Y}}}}\right) + \varepsilon_{\mathcal{X}\leftarrow\mathcal{Y}}. \tag{3}$$

## B.2 STEPWISE DISTILLATION WITH HOMOGENEOUS ANSNET

**Direct distillation.** Training $\mathcal{S}$ directly from $\mathcal{G}$ gives

$$R(\hat{s}) - R(f^\star) \;\leq\; O\!\left(\frac{C_\mathcal{G}}{n^{\rho_{\mathcal{G}\leftarrow f^\star}}} + \frac{C_\mathcal{S}}{n^{\rho_{\mathcal{S}\leftarrow\mathcal{G}}}}\right) + \varepsilon_{\mathcal{G}\leftarrow f^\star} + \varepsilon_{\mathcal{S}\leftarrow\mathcal{G}}. \tag{4}$$

**Stepwise distillation.** Proceeding through $\mathcal{H}_1, \ldots, \mathcal{H}_k$ yields

$$R(\hat{s}) - R(f^\star) \;\leq\; O\!\left(\frac{C_\mathcal{G}}{n^{\rho_{\mathcal{G}\leftarrow f^\star}}} + \sum_{i=1}^{k}\frac{C_{\mathcal{H}_i}}{n^{\rho_{\mathcal{H}_i\leftarrow\mathcal{H}_{i-1}}}} + \frac{C_\mathcal{S}}{n^{\rho_{\mathcal{S}\leftarrow\mathcal{H}_k}}}\right)$$

$$+ \varepsilon_{\mathcal{G}\leftarrow f^\star} + \sum_{i=1}^{k}\varepsilon_{\mathcal{H}_i\leftarrow\mathcal{H}_{i-1}} + \varepsilon_{\mathcal{S}\leftarrow\mathcal{H}_k}, \tag{5}$$

where $\mathcal{H}_0 := \mathcal{G}$.

**Assumption 1** (Rate improvement for homogeneous case). *For all $i$,*

$$\rho_{\mathcal{S}\leftarrow\mathcal{G}} \;\leq\; \min\!\left(\rho_{\mathcal{H}_i\leftarrow\mathcal{H}_{i-1}}, \rho_{\mathcal{S}\leftarrow\mathcal{H}_k}\right).$$

**Assumption 2** (Approximation decomposition for homogeneous case).

$$\sum_{i=1}^{k}\varepsilon_{\mathcal{H}_i\leftarrow\mathcal{H}_{i-1}} + \varepsilon_{\mathcal{S}\leftarrow\mathcal{H}_k} \;\leq\; \varepsilon_{\mathcal{S}\leftarrow\mathcal{G}}.$$

**Conclusion.** Under these assumptions, the multi-step bound equation 5 is asymptotically tighter than the direct bound equation 4, establishing the theoretical advantage of homogeneous stepwise distillation.

## B.3 STEPWISE DISTILLATION WITH HETEROGENEOUS ANSNET

Now consider the case where the original large teacher belongs to a different architecture family $\mathcal{G}^{(A)}$ (e.g., Llama3-8B, Qwen3-4B). We insert a GPT2-XL checkpoint, denoted $\mathcal{G}^{(\mathrm{GPT2})}$, as a bridging assistant before continuing with GPT-2 style checkpoints.

**Additional notation.**

- Mapping operator $\Pi_{\mathrm{map}}$: projects $\mathcal{G}^{(A)}$ outputs into the GPT-2 output space.
- Mapping residual $\delta_{\mathrm{map}}$: additional approximation error from projection mismatch.

The error from $\mathcal{G}^{(A)}$ to $\mathcal{G}^{(\mathrm{GPT2})}$ decomposes as

$$\varepsilon_{\mathcal{G}^{(\mathrm{GPT2})}\leftarrow\mathcal{G}^{(A)}} = \varepsilon_{\mathcal{G}^{(\mathrm{GPT2})}\leftarrow\Pi_{\mathrm{map}}(\mathcal{G}^{(A)})} + \delta_{\mathrm{map}}. \tag{6}$$

**Heterogeneous stepwise bound.** Let $\mathcal{H}_{-1} := \mathcal{G}^{(A)}$ and $\mathcal{H}_0 := \mathcal{G}^{(\mathrm{GPT2})}$, then

$$R(\hat{s}) - R(f^\star) \leq O\!\left(\frac{C_{\mathcal{G}^{(A)}}}{n^{\rho_{\mathcal{G}^{(A)}\leftarrow f^\star}}} + \sum_{i=0}^{k}\frac{C_{\mathcal{H}_i}}{n^{\rho_{\mathcal{H}_i\leftarrow\mathcal{H}_{i-1}}}} + \frac{C_\mathcal{S}}{n^{\rho_{\mathcal{S}\leftarrow\mathcal{H}_k}}}\right)$$

$$+ \varepsilon_{\mathcal{G}^{(A)}\leftarrow f^\star} + \sum_{i=0}^{k}\varepsilon_{\mathcal{H}_i\leftarrow\mathcal{H}_{i-1}} + \varepsilon_{\mathcal{S}\leftarrow\mathcal{H}_k}, \tag{7}$$

where $\varepsilon_{\mathcal{H}_0\leftarrow\mathcal{H}_{-1}}$ includes $\delta_{\mathrm{map}}$.

**Assumption 3** (Rate improvement across mapped chain).

$$\rho_{\mathcal{S}\leftarrow\mathcal{G}^{(A)}} \;\leq\; \min\!\left(\rho_{\mathcal{H}_i\leftarrow\mathcal{H}_{i-1}} \;:\; i = 0, \ldots, k, \rho_{\mathcal{S}\leftarrow\mathcal{H}_k}\right).$$

**Assumption 4** (Approximation decomposition with mapping error).

$$\delta_{\mathrm{map}} + \sum_{i=0}^{k}\varepsilon_{\mathcal{H}_i\leftarrow\mathcal{H}_{i-1}} + \varepsilon_{\mathcal{S}\leftarrow\mathcal{H}_k} \;\leq\; \varepsilon_{\mathcal{S}\leftarrow\mathcal{G}^{(A)}}.$$

**Conclusion.** Under these assumptions, the heterogeneous bound equation 7 is asymptotically tighter than direct distillation from $\mathcal{G}^{(A)}$ to $\mathcal{S}$. This formalizes the role of GPT2-XL as a bridging assistant for cross-architecture stepwise distillation.

## C  EXPERIMENTAL DETAILS

### C.1  TRAINING DETAILS

Our training pipeline consists of four main stages: (1) homogeneous distillation for constructing the **CoL**, (2) heterogeneous distillation, (3) downstream fine-tuning of DesNets, and (4) pre-training of randomly initialized DesNets on OpenWebText.

**Homogeneous Distillation.** We perform homogeneous distillation in two phases to construct the **CoL** using architectures identical to the target model. In the first phase, both the teacher model (AnsNet) and intermediate **CoL** models are supervised-fine-tuned (SFT) on the Dolly dataset for 3 epochs to align model behaviors. In the second phase, we adopt a policy distillation scheme inspired by MiniLLM. Specifically, we employ reverse KL divergence between the teacher and student policies on Dolly, collecting 256 samples per batch and running 4 inner optimization epochs per batch. The clipping parameter $\epsilon$ is set to 0.2, and the sequence length is capped at 512 tokens. During distillation, we sample from the teacher with a temperature of 1. The distillation proceeds for 5000 steps, and we select the final model checkpoint based on Rouge-L score on the validation set.

**Heterogeneous Distillation.** For cross-architecture distillation, we first fine-tune two heterogeneous teacher models—Qwen-4B and LLaMA3-8B—on the Dolly dataset for 3 epochs using a batch size of 2 and a learning rate of 1e-5. We then use the final checkpoints of these teachers to generate pseudo answers given Dolly questions. These synthetic question-answer pairs are used to fine-tune a smaller proxy model (GPT2-XL) for 10 epochs, with a batch size of 8 and learning rate of 1e-5. Subsequently, we distill from the proxy teacher into the **CoL** following the same homogeneous strategy described above.

**DesNet Fine-tuning.** Once the **CoL** is constructed and assembled into target DesNet architectures, we fine-tune the resulting DesNets on downstream tasks for 10 epochs. We use a batch size of 16 or 8 and a learning rate of 1e-5. The model parameters from the final epoch are used for performance evaluation.

**Pre-training from Scratch.** For baseline comparison, we pre-train DesNets from scratch on the OpenWebText corpus. We train for 2 epochs using a learning rate of 5e-4 and a batch size of 2. In this setting, "78M tokens" refers to the full OpenWebText-100K subset (approximately 78M tokens), while "100M", "500M", and "10B tokens" denote the first 100 million, 500 million, and 10 billion tokens extracted sequentially from the full OpenWebText dataset, respectively.

**Fine-tuning Prompts.** Here, we present the prompt templates used in all experiments. For clarity, we abstract away model-specific formatting tokens and show only the semantic content of the prompts. In all experiments, we use the following instruction-style prompts.

For samples containing only the article title, the prompt is:

```
Below is an instruction that describes a task. Write a response that
appropriately completes the request.

### Instruction:
Given the following article, and the title of the article is {instruction},
please summarize the article.

### Response:
```

For samples that additionally contain the full article text, the prompt is:

```
Below is an instruction that describes a task, paired with an input that
provides further context. Write a response that appropriately completes
the request.

### Instruction:
Given the following article, and the title of the article is {instruction},
please summarize the article.
```

```
### Input:
{input}

### Response:
```

## C.2 EVALUATION METRICS

We use Rouge-L to evaluate the performance of our model on generation tasks such as summarization. It measures the longest common subsequence (LCS) between the generated text and the reference summary, which captures sentence-level fluency and coherence better than n-gram-based metrics. Unlike Rouge-n, Rouge-L does not require consecutive matches but considers the in-sequence matches, making it more appropriate for abstractive summarization. The Rouge-L score is defined as the F-measure of the LCS-based precision and recall:

$$\text{Rouge-L} = \frac{(1 + \beta^2) \cdot \text{Precision} \cdot \text{Recall}}{\text{Recall} + \beta^2 \cdot \text{Precision}}, \tag{8}$$

where Precision $= \frac{\text{LCS}(X,Y)}{|X|}$, Recall $= \frac{\text{LCS}(X,Y)}{|Y|}$, $X$ and $Y$ represent the generated and reference summaries respectively, and $\beta$ is typically set to 1.

We also adopt Accuracy (Acc) as the evaluation metric for classification-based tasks such as MMLU and BoolQ. It measures the proportion of correctly predicted instances over the total number of examples. Given a dataset with $N$ samples, the accuracy is computed as:

$$\text{Accuracy} = \frac{1}{N} \sum_{i=1}^{N} \mathbb{I}(\hat{y}_i = y_i), \tag{9}$$

where $\hat{y}_i$ is the predicted label, $y_i$ is the ground-truth label, and $\mathbb{I}(\cdot)$ is the indicator function that returns 1 if the argument is true and 0 otherwise. Note that, in the main paper, we replaced Accuracy (Acc) with Rouge-L to maintain consistency and conciseness across the tables. This substitution is justified as Rouge-L yields results consistent with Accuracy for multiple-choice and true/false questions.

## C.3 HARDWARE

All results reported in this paper are conducted on HUAWEI ASCEND 910B (64G) NPUs.

## D DETAILED INFORMATION OF DATASETS

This section details the information of the datasets adopted in the main paper. As we described in the paper, we benchmark on six datasets: MMLU, English_XLSum, HellaSwag, WinoGrande, BoolQ, and Dolly.

### D.1 MMLU

The Massive Multitask Language Understanding (MMLU) is designed to evaluate the general knowledge and reasoning abilities of language models across 57 diverse subjects, ranging from humanities and social sciences to professional domains. Each question is multiple-choice and intended to reflect educational or professional scenarios. MMLU emphasizes zero-shot and few-shot settings, making it a widely adopted standard for assessing cross-domain generalization in foundation models.

### D.2 ENGLISH_XLSUM

English_XLSum is a single-document summarization dataset comprising 226,711 BBC news articles and corresponding one-sentence summaries. The task focuses on generating concise and abstractive summaries that answer the question "What is this article about?". The dataset covers a broad range of topics and is split into training (204,045), validation (11,332), and test (11,334) sets.

### D.3 HELLASWAG

HellaSwag is a benchmark for commonsense natural language inference, consisting of 70,000 multiple-choice sentence completion problems. While trivial for humans (95%+ accuracy), HellaSwag poses significant challenges for language models due to its adversarial filtering (AF) construction. The dataset was introduced to expose the limitations of deep pre-trained models in commonsense reasoning and has become a cornerstone for evaluating robustness under ambiguous and nuanced scenarios.

### D.4 WINOGRANDE

WinoGrande is a large-scale benchmark inspired by the Winograd Schema Challenge. It consists of 44,000 fill-in-the-blank problems designed to test commonsense reasoning. By scaling up the number of examples and minimizing dataset-specific biases, WinoGrande offers a more robust measure of a model's ability to resolve coreference ambiguities using real-world knowledge.

### D.5 BOOLQ

BoolQ is a yes/no question answering dataset containing 15,942 examples, collected in a naturally occurring, unprompted setting. Each instance includes a question, a passage, and a binary answer. The task setup closely resembles natural language inference and is widely used to evaluate a model's capacity for contextual understanding and factual reasoning.

### D.6 DOLLY

Dolly, developed by Databricks, is an instruction-following language model based on the pythia-12B architecture and fine-tuned on approximately 15,000 instruction-response pairs (databricks-dolly-15k). The data covers diverse task types such as classification, generation, QA, and summarization. While not state-of-the-art, Dolly exhibits strong instruction-following behavior and is fully open-source and commercially licensed, making it a valuable model for experimentation and adaptation in real-world scenarios.

We choose the **Dolly** dataset as the training corpus for constructing the **CoL** due to its broad coverage of language capabilities and proven effectiveness in previous distillation research such as MiniLLM. Dolly spans a diverse range of task types, including brainstorming, classification, closed-form QA, open-ended QA, summarization, generation, and information extraction. This diversity enables **CoL** to capture reusable knowledge blocks that generalize across different task formats. In contrast, most other datasets such as MMLU, BoolQ, and English_XLSum focus on narrower task types—e.g., closed QA or summarization—which may limit the diversity of distilled knowledge. Leveraging a multi-capability dataset like Dolly is thus critical for building a more general and robust **CoL**.

## E DETAILED ARCHITECTURES OF DESNETS

In this section, we list the detailed information of the DesNets shown in the main paper. Specifically, we show the number of layers, multi-head attentions, hidden dimensions and the intermediate dimensions of the FFN layers. The structure of each DesNet is described in Table 7.

## F DETAILED VALUES

Here, we show the detailed values of Figure 3 in the main paper, as shown in Table 8. The detailed results further validate our view that LInit can effectively save the training corpus required for pre-training, regardless of whether the AnsNet are homogeneous (GPT2-XL) or heterogeneous (Llama3-8B, Qwen3-4B).

Table 7: The detailed structure of the DesNets in the main paper. 'P' denotes the parameters of the DesNet, 'He$_{Dim}$' is the dimension of each multi-head attention, 'Hidden$_{Dim}$' means the dimension of the hidden size, and 'I$_{Dim}$' is the intermediate dimension.

| P(M) | Layer | Head | He$_{Dim}$ | Hidden$_{Dim}$ | I$_{Dim}$ |
|---|---|---|---|---|---|
| 138 | 14 | 12 | 64 | 768 | 3072 |
| 220 | 18 | 14 | 64 | 896 | 3584 |
| 277 | 24 | 14 | 64 | 896 | 3584 |
| 380 | 26 | 16 | 64 | 1024 | 4096 |
| 537 | 30 | 18 | 64 | 1152 | 4608 |

Table 8: The detailed values of Figure 3 in the main paper.

| Task | CoL | | | Rand | | | | | |
|---|---|---|---|---|---|---|---|---|---|
| | GPT2-XL | Llama3-8B | Qwen3-4B | 0.078B | 0.1B | 0.5B | 1B | 5B | 10B |
| DesNet-138M | | | | | | | | | |
| English_XLsum | 32.19 | 33.57 | 31.35 | 26.18 | 26.95 | 31.28 | 30.71 | 33.52 | 35.26 |
| HellaSwag | 30.59 | 39.59 | 42.01 | 23.68 | 23.68 | 29.47 | 31.81 | 29.67 | 29.27 |
| WinoGrande | 63.20 | 53.72 | 60.20 | 51.50 | 52.50 | 54.80 | 56.80 | 56.80 | 55.40 |
| DesNet-380M | | | | | | | | | |
| English_XLsum | 39.29 | 37.15 | 36.31 | 22.20 | 22.62 | 31.45 | 34.40 | 35.87 | 40.06 |
| HellaSwag | 47.47 | 35.90 | 46.64 | 38.54 | 36.96 | 41.00 | 40.71 | 45.93 | 46.55 |
| WinoGrande | 68.00 | 49.48 | 67.30 | 36.90 | 35.30 | 51.10 | 58.90 | 59.20 | 59.40 |

## G  THE ADDITIONAL COMPARISON OF CONVERGENCE SPEED

In this section, we first compare the convergence speed of DesNet-138M initialized by **CoL** (LInit) and from scratch (Rand) on 100M pre-training tokens across three AnsNet: GPT2-XL, Llama3-8B, and Qwen3-4B, as shown in Figure 7. Then, we show the the convergence speed of DesNet-138M initialized by **CoL** (LInit) and from scratch (Rand) on 500M, 10B pre-training tokens, and the results are shown in Figure 8. Finally, we further show the convergence speed of DesNet-220M, DesNet-277M, DesNet-380M, and DesNet-537M initialized by **CoL** (LInit) and from scratch (Rand) on 78M, 100M, 500M, 10B pre-training tokens across the AnsNet: GPT2-XL. Both results are shown in Figure 9.

From the both results, we can obviously observe that LInit has faster convergence speed than pre-training from scratch. For example, LInit achieves the same loss in only 140 and 5000 steps, compared to 30500, and 250200 steps for Pre on 500M, and 10B tokens, respectively, leading to a speedup of 80.26×, 217.86×, and 50.04×, as shown in Figure 8. Besides, for DesNet-220M, LInit reaches the same loss level using 1350, 1700, 2500, and 7350 steps under 78M, 100M, 500M, and 10B token budgets, respectively, while Pre requires 4750, 6100, 11000, and 31150 steps, as shown in Figure 9. This corresponds to a convergence speedup of 3.52×, 3.59×, 4.4×, and 4.24×. These results demonstrate that LInit consistently accelerates convergence across different token budgets and model sizes, with particularly significant gains in the low-token regime.

## H  PERFORMANCE GAP WITH THE INCREASE OF THE PRE-TRAINING TOKEN

Figure 10 illustrates the performance change as the number of pre-training tokens increases. All DesNets are initialized from the AnsNet GPT2-XL. The results show that the performance of pre-training from scratch (Rand) improves significantly with more pre-training data. This improvement is particularly substantial for DesNets with 380M and 537M parameters when the pre-training data reaches 10B tokens. In contrast, the LInit method proposed in this paper maintains outstanding performance under the same number of tokens. These results further demonstrate the superiority of the proposed LInit method.

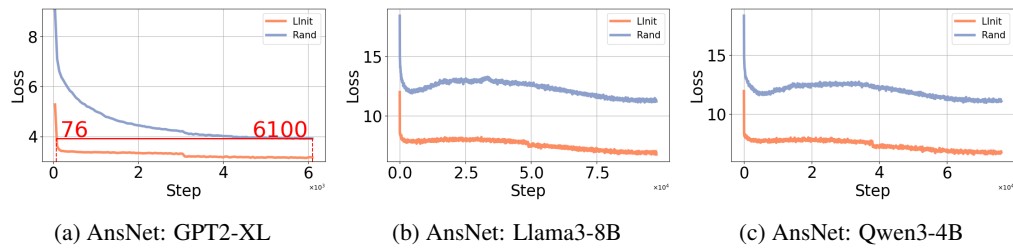

Figure 7: Convergence speed of DesNet-138M initialized by **CoL** (LInit) and from scratch (Rand) on 100M pre-training tokens across three AnsNet.

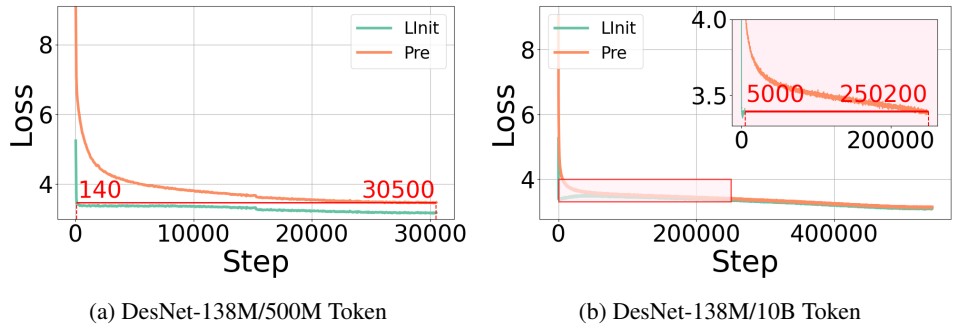

Figure 8: Convergence speed of DesNet-138M initialized by **CoL** (LInit) and from scratch (Rand) on 500M and 10B pre-training tokens across the AnsNet: GPT2-XL.

## I PERFORMANCE COMPARISON BETWEEN THE INITIALIZATION WAY FROM CoL AND THE PREVIOUS LEARNGENE METHOD

In this section, we compare the proposed initialization way from the **CoL** against the representative previous learngene method, such as Auto-Learngene (Wang et al., 2023b), and Vanilla-Learngene (Wang et al., 2022a) methods.

Auto-Learngene proposes to automatically identify key layers of AnsNet for Transformer architectures by introducing a MetaNet to guide the selection process. In experiments, Auto-Learngene observes that the lower layers of Transformers usually exhibit stronger generalization ability. Therefore, it directly selects the first N layers of the Transformer and stacks randomly initialized layers on top to construct DesNets with different architectures. In contrast, Van-Learngene identifies key layers of AnsNet in CNNs based on gradient signals. Empirically, Van-Learngene finds that the higher layers of CNNs tend to contain richer semantic information, and thus it extracts the last three layers as learngene and stacks N randomly initialized layers before them to build multiple DesNets.

Since neither of these approaches was originally designed for LLMs, we migrate them into the LLM setting. Specifically, for Auto-Learngene, we extract the first three layers of AnsNet as learngene and stack 0, 2, or 4 randomly initialized layers after them to construct three descendant models of different sizes. For Van-Learngene, we extract the last three layers of AnsNet as learngene and stack 0, 2, or 4 randomly initialized layers before them to construct three descendant models of different sizes. We use GPT2-XL as AnsNet, because if Qwen3-4B or Llama3-8B were used, even extracting only one layer as learngene without stacking any random layers would already result in descendant models with over 500M and 1.4B parameters, respectively, which are unsuitable for our small language model setting.

In addition, there exist other Learngene-based methods as introduced in the related work, such as Learngene Pool and SWS. These methods are both based on direct distillation with auxiliary models. In the ViT domain this strategy works because the capacity gaps are small, for example, DeiT-Base/ViT-Base ( 86M) is distilled into 3M or 44M auxiliary models. However, directly transferring such designs to LLMs is infeasible. We fail to transfer those two methods into LLM, and the perfor-

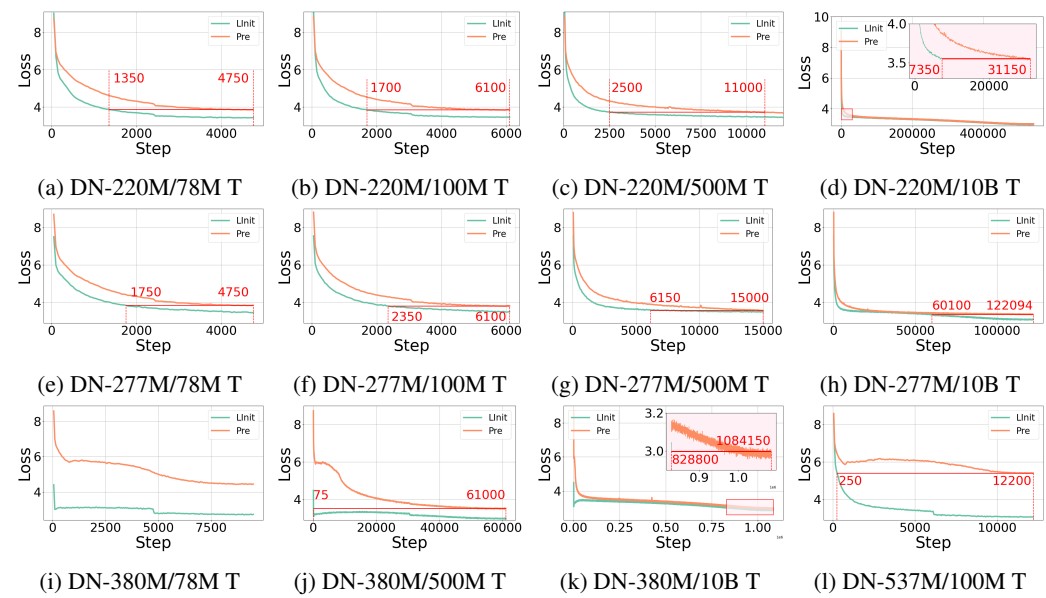

Figure 9: Convergence speed of variable-sized DesNets initialized by **CoL** (LInit) and from scratch (Rand) on varying pre-training tokens across the AnsNet: GPT2-XL. 'DN' denotes DesNet, and 'T' means Token.

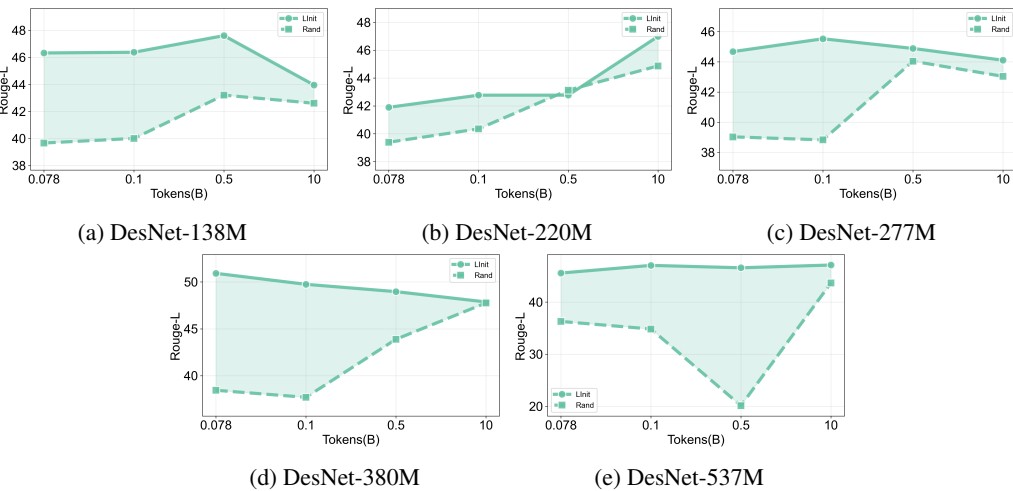

Figure 10: Performance gap between the LInit and random initialization (Rand) on the Dolly task with the changing of the pretrain tokens.

mance of the initialized DesNets is even worse than that of the random initialization. This is due to the large gap between LLMs and small LMs. Moreover, they typically rely on multi-model distillation or staged expansion to form a learngene pool or modular structure. Such designs significantly increase computational costs and are difficult to scale to large LLMs. Therefore, in this study, we select Auto-Learngene and Van-Learngene as the most representative baseline methods to migrate and compare, as they both capture the core principles of learngene while aligning with our research objectives.

As shown in Table 9, we can find that the initialization method from the proposed method achieves the best results on the average score across three scales of the DesNet. Furthermore, for specific tasks, **CoL** achieved the best results in the vast majority of them, and the result values were far higher than those of the previous Learngene methods. Specifically, when the scale of the descendant model was 138M, **CoL** achieved a result of 62% on WinoGrande. In contrast, auto-learngene and

Table 9: Performance comparison between the proposed initialization way from **CoL** and representative previous learngene methods.

| Method | Params | Dolly | BoolQ | HellaSwag | MMLU | WinoGrande | English_XLSum | Avg |
|---|---|---|---|---|---|---|---|---|
| Auto-Learngene | 174.28 | 20.30 | 65.65 | 34.98 | 25.47 | 44.72 | **33.91** | 37.50 |
| Van-Learngene | 174.28 | 19.62 | 63.79 | **36.24** | 24.62 | 41.83 | 32.90 | 36.50 |
| **CoL** | 134.00 | **22.99** | **76.80** | 31.15 | **31.00** | **62.00** | 30.68 | **42.44** |
| Auto-Learngene | 235.76 | **20.03** | 60.88 | **36.45** | 24.66 | 44.12 | **35.18** | 36.89 |
| Van-Learngene | 235.76 | 19.92 | 55.86 | 35.85 | 24.46 | 42.23 | 33.30 | 35.27 |
| **CoL** | 220.00 | 19.82 | **69.50** | 26.84 | **28.50** | **58.00** | 26.68 | **38.22** |
| Auto-Learngene | 297.24 | 20.43 | 61.94 | 35.02 | 24.53 | 44.28 | 31.86 | 36.34 |
| Van-Learngene | 297.24 | 20.67 | 62.62 | 34.28 | 23.77 | 44.48 | **32.70** | 36.42 |
| **CoL** | 277.00 | **20.82** | **71.80** | **43.03** | **28.20** | **61.49** | 32.36 | **42.95** |

van-learngene only achieved results of 44.72% and 41.83% respectively. **CoL** outperformed them by 17.28% and 20.17%. This demonstrates the superiority of the **CoL** method.

## J  RESULTS OF QWEN3-BASED **CoL**

To further verify the generality of the proposed **CoL** methods, we conduct the experiments on Qwen3 Yang et al. (2025) family. Here, we take Qwen3-4B as the AnsNet, and the learngne chain contains Qwen3-1.7B, Qwen3-0.6B. The DesNet built from the Qwen3-based learngene chain has 1.07B parameters, and we compare it with random initialization. The results are shown in Table 10, and we find that Qwen3-based **CoL** still outperforms the random initialization on many tasks. This demostrates that the presented results and trends hold for more recent models like Qwen3.

Table 10: Performance comparison between the Qwen3-based**CoL** and Random initialization on DesNet-1.09B.

| | BoolQ | Arc-E | WinoG | Avg |
|---|---|---|---|---|
| Random | 64.94 | 25.26 | 43.80 | 44.67 |
| CoL | 65.76 | 25.44 | 44.44 | **45.21** |

## K  COMPARISON WITH HYPERCLONING AND FSLM

In this section, we provide additional analysis comparing **CoL** with two related works, Hyper-Cloning Samragh et al. (2024) and Stacked Small Language Models (FSLM) Liang (2024).

**The difference between CoL and HyperCloning, FSLM**. Although these methods share the broad goal of enhancing small language models (SLMs), their objectives, system designs, and scalability properties differ fundamentally from **CoL**. HyperCloning focuses on accelerating the training of a *single* large model by progressively widening its architecture. FSLM constructs a multi-component system composed of several SLMs, but it does not offer a mechanism to initialize a large number of descendant models (DesNets) at multiple scales. In contrast, **CoL** is explicitly designed to generate and initialize *multiple* SLMs of different sizes under strict compute and data budgets.

**Empirical Comparison with HyperCloning**. To make the comparison favorable to HyperCloning, we follow its standard widening schedule and initialize the HyperCloning model from a strong **CoL**-initialized DesNet-138M, rather than from random initialization. Using 500M pretraining tokens, we obtain HyperCloning models at two scales: 645M and 2.42B parameters. **CoL** descendants are constructed at comparable scales, and the results are shown in Table 11.

Across all fairness categories, **CoL** consistently outperforms HyperCloning, even when trained with significantly fewer tokens. This highlights **CoL**'s efficiency and its ability to transfer useful knowledge with minimal computation.

Table 11: Downstream performance comparison between HyperCloning and **CoL** under different scaling and data regimes.

| Line | Method | Tokens | Param | BoolQ | MMLU | WinoG | Avg |
|------|--------|--------|-------|-------|------|-------|-----|
| 1 | HyperCloning | 500M | 2423M | 63.55 | 24.21 | 45.88 | 38.28 |
| 2 | HyperCloning | 500M | 645M | 63.90 | 23.99 | 46.47 | 38.00 |
| 3 | CoL | 500M | 277M | 67.80 | 27.10 | 63.74 | 43.87 |
| 4 | CoL | 500M | 537M | 67.80 | 25.30 | 62.30 | 43.69 |
| 5 | CoL | 100M | 537M | 71.90 | 26.30 | 60.20 | 44.06 |

**Comparison with FSLM.** We additionally compare **CoL** with the official results of FSLM (Stacked Small Language Models). FSLM produces a 640M-parameter multi-SLM system using 5K supervised training samples. In contrast, **CoL** generates a single 138M DesNet (only 21.25% of FSLM's size) with approximately half as many SFT samples (2420 vs. 5000).

Table 12: Comparison between FSLM and **CoL** under strict resource constraints.

| Method | SFT Samples (K) | Param (M) | TinyARC/ARC | TinyMMLU/MMLU |
|--------|-----------------|-----------|-------------|----------------|
| FSLM | 5.0 | 640 | 33.49 | 32.08 |
| CoL | 2.139+0.285 | 138 (21.25%) | 29.50 (88.08%) | 28.20 (87.91%) |

Even under such a large scale and data disadvantage, **CoL** achieves **88.08%** of FSLM's performance on TinyARC and **87.91%** on TinyMMLU. This shows that FSLM is not well suited for resource-limited scenarios, while **CoL** provides strong initialization quality even at very small model sizes.

Overall, these findings demonstrate that HyperCloning and FSLM differ fundamentally from **CoL** in goals and scalability, and **CoL** provides superior performance, faster convergence, and substantially better resource efficiency across a range of evaluation settings.

## L  CoL PERFORMANCE ACROSS EXTREME MODEL SCALES

To examine the applicability limits of **CoL**, we evaluate its performance on two extreme model-size regimes: very small SLMs (<50M parameters), where representational capacity is highly constrained, and (2) larger-than-target models (>1B parameters), where the available token budget becomes insufficient for meaningful training.

### L.1  VERY SMALL MODELS (<50M)

We construct a 51.47M-parameter descendant model (DesNet-51.47M) and compare **CoL** initialization with random initialization under an identical 100M-token training budget. Table 13 shows that **CoL** consistently outperforms random initialization across all downstream tasks, despite the extremely limited model capacity.

Table 13: Performance of DesNet-51.47M initialized with **CoL** and random initialization under the same 100M-token budget.

| Method | BoolQ | MMLU | WinoG | XSum | HellaS | Arc-E | Avg |
|--------|-------|------|-------|------|--------|-------|-----|
| Rand | 51.55 | 24.29 | 41.23 | 33.11 | 35.56 | 27.36 | 35.52 |
| CoL | 65.25 | 24.20 | 52.42 | 34.91 | 39.05 | 25.82 | 40.27 |

These results confirm that **CoL** remains effective even in the extremely small-model regime, providing substantial performance gains over random initialization.

## L.2 LARGE MODELS (>1B)

We explore whether CoL can scale to DesNets larger than 1B parameters. With the GPT2-based learngene chain, CoL becomes difficult to apply at this scale. The main reason is simple: the largest checkpoint in the GPT2 chain is only 734M. This checkpoint is too small to serve as an effective AnsNet for training a 1B-level DesNet. As a result, both random initialization and CoL show poor convergence under a 100M-token budget.

To test whether this limitation comes from the chain rather than CoL itself, we tried a Qwen3-based learngene chain. Its largest checkpoint is 1.7B, which is large enough to act as a suitable AnsNet. Within this chain, the DesNet-1.09B can achieves better performance than random initialization under the 100M-token budget, as shown in Table 10. These results show that scaling CoL beyond 1B depends on the size of the AnsNet in the chain. CoL remains effective, but its scalability is constrained by the maximum checkpoint size in the learngene chain.

## M THE ROLE OF BRIDGE DISTILLATION

We compare heterogeneous bridge distillation with the homogeneous GPT2-based CoL chain on several generation-style benchmarks, including dolly (Conover et al., 2023), self-inst Wang et al. (2023c), sinst Wang et al. (2022c), uinst Honovich et al. (2023), and vicuna Chiang et al. (2023). The results for DesNet-138M and DesNet-380M are reported in Table 14 and Table 15. Across all tasks and both model sizes, heterogeneous paths (Qwen3-4B or Llama3-8B as the AnsNet) achieve consistently higher scores than the GPT2-XL CoL chain.

These results differ from the multiple-choice benchmarks in the main paper, where homogeneous chains perform better. This contrast reflects the intrinsic properties of different task types. Multiple-choice tasks benefit more from architectural homogeneity and token-level alignment, whereas open-ended generation tasks rely more on high-level semantic priors that large modern AnsNets provide. Thus, heterogeneous bridge distillation is not weaker; it offers clear advantages in generative settings and highlights the task-dependent behavior of CoL.

Table 14: Performance of bridge distillation on generation tasks for DesNet-138M.

| AnsNet | dolly | self-inst | sinst | vicuna | Avg |
|---|---|---|---|---|---|
| GPT2-XL | 22.10 | 9.35 | 14.26 | 15.31 | 12.97 |
| Qwen3-4B | 25.25 | 11.79 | 15.64 | 15.53 | **14.32** |
| Llama3-8B | 22.83 | 10.99 | 15.48 | 15.86 | 14.11 |

Table 15: Performance of bridge distillation on generation tasks for DesNet-380M.

| AnsNet | dolly | self-inst | sinst | vicuna | Avg |
|---|---|---|---|---|---|
| GPT2-XL | 26.20 | 11.01 | 16.90 | 15.77 | 17.47 |
| Qwen3-4B | 27.55 | 13.58 | 21.51 | 17.49 | **20.03** |
| Llama3-8B | 26.63 | 12.95 | 21.65 | 16.75 | 19.49 |

## N THE USE OF LLMS

In this paper, our application of LLM mainly involves three aspects. Firstly, we use LLM to correct and polish the words and grammar in the sentences of the paper. Subsequently, we used LLM to beautify the graphs in the experiment. Specifically, we use LLM to modify the existing drawing python code to achieve the purpose of beautification. Finally, we use LLM to assist in checking the formulas and theoretical derivation processes in the paper. We declare that all the generated content of LLMS has been reviewed by the authors.

## O   LIMITATIONS AND FUTURE WORK

**Limitations.** In this work, we demonstrated the effectiveness of the proposed **CoL** framework primarily through averaged performance across multiple datasets. While **CoL**consistently achieves strong overall results, it does not yield the best performance on every individual task. This limitation arises because **CoL** emphasizes task-agnostic knowledge transfer and does not explicitly incorporate task-specific adaptations, which may lead to suboptimal outcomes in certain cases.

**Future Work.** A promising direction for future research is to enhance **CoL** with task-aware initialization strategies. By incorporating characteristics of the downstream application scenarios, **CoL** could selectively adapt and initialize parameters that are most sensitive to the target task. Such extensions have the potential to yield more comprehensive improvements in the performance of SLMs on specific tasks, further broadening the applicability of **CoL**.

