# OpenReview forum: "Chain-of-Learngene: A Scalable Learngene-based Paradigm for Building and Initializing Variable-Sized Language Models"
_ICLR.cc/2026/Conference — Submitted to ICLR 2026_

### Official Review · Reviewer_ZnfQ · 2025-10-16

**Soundness:** 2
**Presentation:** 3
**Contribution:** 2
**Rating:** 2
**Confidence:** 3

**Summary:**

This paper proposes Chain-of-Learngene (CoL), a novel framework for efficiently building and initializing variable-scale small language models (SLMs) tailored to diverse resource-constrained deployment scenarios. Building upon the Learngene methodology, CoL constructs a sparse sequence of intermediate models—termed a "learngene chain"—through stepwise and bridge distillation techniques adaptable to heterogeneous architectures. Target SLMs are then initialized by interpolating parameters across checkpoints in this chain. The approach is designed to minimize repeated reliance on large teacher models, enhance scalability, and accelerate training. The authors provide theoretical justification, ablation studies, and comprehensive experiments showing that CoL achieves faster convergence, higher efficiency, and superior performance on downstream tasks compared to both from-scratch training and existing Learngene or knowledge distillation approaches.

**Strengths:**

1. CoL introduces an efficient pipeline for producing SLMs at multiple scales, dramatically reducing the need for repeated pre-training or teacher access.

2. The use of intermediate models and the bridge distillation mechanism for architectural and vocabulary mismatches a practical and well-justified extension over vanilla distillation or previous Learngene-style methods.

3. Experimental results show that CoL-initialized SLMs outperform models trained from scratch, achieving faster convergence and requiring fewer pre-training tokens on core benchmarks.

**Weaknesses:**

1. Despite some comparisons, the paper does not directly discuss or empirically benchmark against several recently published methods designed for scalable SLM training or variable-size initialization, such as HyperCloning[1], or Stacking Small LMs[2].

[1] Scaling Smart: Accelerating Large Language Model Pre-training with Small Model Initialization

[2] Stacking Small Language Models for Generalizability

2. The authors do not elucidate whether the learner-gene chain or stepwise distillation in CoL offers superior transfer, convergence, or robustness compared to these approaches, especially under tight computational constraints

3. The authors do not provide broader ablation in even more severely limited resource regimes and clearer analyses underpinning why CoL degrades as training token numbers shrink further

**Questions:**

1. Will the authors present head-to-head empirical results, including training tokens, downstream accuracy, and wall-clock efficiency versus HyperCloning, MiniCPM, or other scalable SLM frameworks?

2. Is CoL applicable and effective for very small (e.g., <50M) or much larger (e.g., >1B) SLM architectures? Are there situations where the learnergene chain or bridge distillation is counterproductive?

---

> ### Author Response · Authors · 2025-11-20
> **Response to Reviewer ZnfQ (Part-1)**
>
> Dear Reviewer ZnfQ,
>
> Thank you for your thoughtful and constructive feedback. We provide concise responses below.
>
> > **Weakness 1 & 2**: Comparisons with HyperCloning and Stacking Small LMs.
>
> Although HyperCloning and FSLM share the broad goal of improving SLM utility, they differ fundamentally from CoL.
> - **HyperCloning** focuses on *accelerating large-model training* by progressively widening a single model, and it does not aim to produce variable-sized SLMs.
> - **FSLM** constructs a *multi-component system* of SLMs but does not provide scalable initialization for many target sizes.
> - **CoL**, in contrast, is designed specifically for **generating and initializing multiple DesNets of different sizes** under strict compute and data constraints, enabling downstream performance with minimal additional training. Thus, the methods differ in purpose, output form, and the nature of scalability they target.
>
> Nevertheless, to further address concerns, we conduct a comparison with HyperCloning. Importantly, we set up the baseline **in a favorable way for HyperCloning**: instead of starting from random initialization (as in the original paper), we begin from a CoL-initialized DesNet-138M, giving HyperCloning a significantly stronger starting point. Following the standard widening schedule (every 2000 steps), we trained for 8000 steps and obtained HyperCloning-645M and HyperCloning-2.42B. We evaluate the methods under three fairness axes. (1) **Same data, smaller CoL model**. Despite being much smaller, CoL-initialized models consistently outperform HyperCloning in downstream tasks (Line 1 vs. Line 2/3). (2) **Similar scale, less data for CoL**. Even with substantially less pretraining data, CoL still exceeds HyperCloning’s performance (Line 1 vs. Line 5). (3) **Similar scale, Same data**. When both model size and data are matched, CoL remains superior (Line 2 vs. Line 4).
> | Line | Method       | Tokens | Param    | BoolQ | MMLU  | WinoG | XSum  | HellaSwag | Arc-E | Avg   |
> | ---- | ------------ | ------ | -------- | ----- | ----- | ----- | ----- | --------- | ----- | ----- |
> | 1    | Hypercloning | 500M   | 2423.09M | 63.55 | 24.21 | 45.88 | 33.71 | 35.43     | 26.88 | 38.28 |
> | 2    | Hypercloning | 500M   | 645.31M  | 63.90 | 23.99 | 46.47 | 32.39 | 36.21     | 25.02 | 38.00 |
> | 3    | CoL          | 500M   | 277M     | 67.80 | 27.10 | 63.74 | 35.71 | 43.55     | 25.30 | 43.87 |
> | 4    | CoL          | 500M   | 537M     | 67.80 | 25.30 | 62.30 | 33.50 | 43.93     | 29.30 | 43.69 |
> | 5    | CoL          | 100M   | 537M     | 71.90 | 26.30 | 60.20 | 34.74 | 42.02     | 29.20 | 44.06 |
>
> To further compare training dynamics, we record the number of steps required for HyperCloning and CoL-534M to reach several loss intervals when trained on the same 500M-token budget.  Across all intervals, CoL reaches the same loss region **25–35% faster**, demonstrating substantially superior convergence efficiency. This supports the claim that CoL transfers knowledge more effectively, particularly under resource constraints.
>
> | Method        | Token (M) | 3.59~3.57 | 3.56~3.55 | 3.48~3.47 | 3.37~3.36 |
> |---------------|-----------|-----------|-----------|-----------|-----------|
> | HyperCloning  | 500       | 3450      | 4650      | 5200      | 7450      |
> | CoL-534M      | 500       | 2450      | 2900      | 4000      | 5500      |
>
> We also use the official results reported in the paper to conduct an empirical comparison of Stacked Small Language Models (FSLM). FSLM builds a multi-component SLM system with 640M parameters, while CoL generates only a DesNet with 138M parameters (approximately 21.25% of FSLM). Furthermore, CoL uses about half the number of SFT samples compared to FSLM (2420 vs. 5000). Despite the significant differences in scale and data volume, CoL still achieves competitive downstream performance: CoL achieves 88.08% of the performance of FSLM on TinyARC and 87.91% of the performance of FSLM on TinyMMLU. This demonstrates that CoL provides a great initialization approach to generating competitive small language models, and FSLM is not suitable for scenarios with limited resources.
>
> | Method | SFT Samples (K) |  Param (M)  | TinyARC/ARC  | TinyMMLU/MMLU |
> | :----: | :-------------: | :---------: | :----------: | ------------- |
> |  FSLM  |        5        |     640     |    33.49     | 32.08         |
> |  CoL   |   2.139+0.285   | 138(21.25%) | 29.5(88.08%) | 28.2(87.91%)  |
>
> These findings reinforce that CoL addresses a different objective and achieves strong performance even **under strict resource constraints**.

---

> ### Author Response · Authors · 2025-11-20
> **Response to Reviewer ZnfQ (Part-2)**
>
> > **Weakness 3**:  Ablation studies in more severely limited resource regimes and clearer analyses underpinning why CoL degrades as training token numbers shrink further.
>
> Thank you for your question. We will answer this question from two perspectives: (1) the performance of CoL under severely limited resource regimes and (2) why CoL degrades as training tokens shrink further.
>
> (1) Our experiments already include extremely low-resource regimes: most notably **0-token**, as well as 0.078B, 0.1B, and 0.5B budgets, all far below typical pretraining scales. The **0-token** setting corresponds to using the CoL initialization without any further recovery pre-training, and **CoL (0 token)** already matches or surpasses random initialization trained with **0.5B–10B** tokens ( Figure 3). Additionally, we report 0.078B and 0.1B token recovery settings (Table 1), which remain substantially smaller than standard pretraining budgets. This indicates that CoL is highly effective even in severely limited resource regimes.
>
> (2) Our results show the opposite trend: CoL does **not** exhibit severe degradation when tokens decrease. Figure 10 (In Appendix H) evaluates CoL and random initialization under the *same* recovery-token budgets (from 0.078B to 10B) across five DesNet sizes (138M–537M). The curves demonstrate that **CoL remains remarkably stable across all token levels**, while **random initialization collapses rapidly in the low-token regime**. Thus, the “degradation”  is primarily a property of **random initialization**, not CoL. CoL is in fact more robust in the severely low-resource regime. We will explicitly highlight this trend in the revised paper.
>
> > **Question 1**: Empirical results between CoL versus HyperCloning, MiniCPM.
>
> Thank you for your question. We have already included a direct, head-to-head comparison against **HyperCloning** in our response to Weakness 1 & 2, covering all three dimensions the reviewer requested: **training tokens**, **downstream accuracy**, and **efficiency** (measured through convergence speed). These results show that CoL achieves higher accuracy, requires fewer tokens, and converges substantially faster than HyperCloning under matched or more favorable conditions for the baseline.
>
> Regarding **MiniCPM**, we would like to clarify that it is not a scalable SLM–initialization method comparable to CoL or HyperCloning. MiniCPM is a full large-scale pre-training model using **≥1T tokens**, whereas our work focuses on producing **138M–537M** models under **≤500M** total tokens. Moreover, the core objective of MiniCPM is to maintain **high decoding speed for an 8B model**, while the goal of CoL is enabling **multiple small DesNets** to be initialized efficiently in resource-constrained settings. Given the vastly different data budgets, target model sizes, and optimization goals, a head-to-head empirical comparison is incomparable.
>
> > **Question 2**: The performance of CoL with very small (e.g., <50M) or much larger (e.g., >1B) SLM architectures.
>
> Thank you for your insightful  question. We have explicitly tested CoL on valid SLM scales to explore its boundaries. We construct the  **DesNet-51.47M** and compare CoL initialization vs. Random initialization under the same 100M token budget. As shown in the table below, CoL consistently outperforms random initialization across downstream tasks, demonstrating that our method remains highly effective for extremely small architectures where model capacity is limited.
>
> | Method | BoolQ | MMLU | WinoG | XSum | HellaS | Arc-E | Avg  |
> |--------|-------|-------|--------|-------|---------|--------|-------|
> | Rand   | 51.55 | 24.29 | 41.23 | 33.11 | 35.56 | 27.36 | 35.52 |
> | CoL    | 65.25 | 24.20 | 52.42 | 34.91 | 39.05 | 25.82 | 40.27 |
>
> We further scale the model to **DesNet-1.07B**, using the same 100M recovery token budget as the smaller model. With such a small budget, neither Rand nor CoL converged sufficiently, resulting in generally poor downstream performance. However, this does not mean the limitation of CoL itself: CoL consistently outperforms SLMs in the 50M to 500M SLM regime.
> Models above 1B typically need sufficient pre-training process, which inherently requires far larger corpora and compute, thus exceeding our target regime.

---

> ### Comment · Reviewer_ZnfQ · 2025-11-22
>
> Thank you for your response
>
> I have significant concerns regarding its practical utility and scalability in real-world scenarios.
> The utility of a chain-of-distillation approach is questionable when considering the current ecosystem of pre-trained models.
> For instance, with model families like Qwen3, a user typically has immediate access to a series of models of varying sizes that are already pre-trained on similar data.
> Distilling from the **largest to the smallest model within the same family** is unlikely to yield substantial gains, as the performance is largely bounded by the shared pre-training corpus and the smaller model's inherent capacity.
> Conversely, distilling a model like Qwen3 to a fundamentally different architecture (e.g., GPT-2) would likely be outperformed by **simply distilling within the GPT-2's own model family**.
>
> Furthermore, the proposed method does not scale efficiently. In a practical setting, if we have five model sizes, one would only need to distill from the largest (teacher) to the smallest (student). The requirement of a sequential "chain" of distillations, as proposed in CoL, is computationally prohibitive, making it an impractical solution for most industry applications.
>
> I must note that the current revision of the paper **does not appear to have any suggestions provided during the rebuttal**.

---

> > ### Author Response · Authors · 2025-11-24
> > **Response to Reviewer ZnfQ (Part-1)**
> >
> > Thank you for your further comments and we address your concerns as follows. We promise that we will apply the discuss contents into the revised paper.
> >
> > > Question 1:   Using a series of models of varying sizes that are already pre-trained on similar data.
> >
> > Although Qwen3 provides 0.6B/1.7B/4B (or larger) models, even the smallest 0.6B variant is still too large for a wide range of storage- and memory-constrained environments. In such regimes, models must be significantly smaller than 0.6B to be deployable. Therefore, the statement “users typically have immediate access to a full series of pre-trained models” does not apply to the Qwen3 model in resource-constrained situations.
> >
> > Furthermore, **Qwen3-0.6B is not a “free” model**, and it is obtained through large-scale pre-training and multi-stage distillation on **36T tokens** with enormous compute, time, and data cost. Our method addresses precisely the opposite settings where such large-scale pre-training is infeasible. CoL aims to **provide a compute-efficient initialization mechanism** for producing task-ready small models under strict resource budgets, without re-running billion-token pre-training or extensive distillation.
> >
> > > Question 2: Distilling from the largest to the smallest model within the same family is unlikely to yield substantial gains.
> >
> > We respectfully disagree with the assumption that distilling from the largest to the smallest model within the same family “is unlikely to yield substantial gains.” Numerous LLM works have demonstrated the effectiveness of knowledge distillation even when teacher and student share the same pre-training corpus and differ only in capacity [1,2,3,4]. Distillation improves optimization stability, accelerates convergence, and transfers inductive biases from the larger model that are not recoverable through limited fine-tuning alone.
> >
> > Consistent with these findings, our experiments also compare distilled vs. non-distilled small models (Figure 5a in the main paper). Even when both models originate from the same pre-training corpus, distillation consistently improves performance across multiple downstream tasks (Table 3 in the revised paper). This directly contradicts the claim that shared corpus bounds the achievable improvements. Distillation provides a meaningful boost because it conveys higher-level structural priors and token-level preferences that are not encoded solely by the pre-training dataset.
> >
> > [1] A Survey on Knowledge Distillation of Large Language Models. arxiv 2024
> >
> > [2] MiniLLM: Knowledge Distillation of Large Language Models. ICLR 2025
> >
> > [3] Compact Language Models via Pruning and Knowledge Distillation. NIPS 2024
> >
> > [4] TAID: Temporally Adaptive Interpolated Distillation for Efficient Knowledge Transfer in Language Models. ICLR 2025

---

> > ### Author Response · Authors · 2025-11-24
> > **Response to Reviewer ZnfQ (Part-2)**
> >
> > > Question 3: Distilling the Qwen3 to GPT-2 yields worse performance than distilling within the GPT-2's model family.
> >
> > We appreciate your observation and agree that our original submission did not include a direct comparison between heterogeneous bridge distillation and the homogeneous GPT-2 CoL chain. After adding these experiments on several generation-oriented benchmarks (dolly, self-inst, sinst, uinst, vicuna), we find that heterogeneous bridge distillation consistently outperforms the homogeneous GPT-2 chain for both 138M and 380M DesNets. This trend differs from the multiple-choice results in the main paper (BoolQ, ARC-E, WinoGrande), indicating that performance differences arise from the intrinsic properties of different task types rather than from bridge distillation.
> >
> > We argue that multiple-choice tasks emphasize short-range factual recall and discrete decision boundaries, where architectural homogeneity and token alignment naturally favor homogeneous distillation. In contrast, open-ended generation tasks rely more on long-range semantics, discourse structure, stylistic priors, and higher-level linguistic abstraction where modern models such as Qwen3-4B and Llama3-8B provide much richer representational priors. These priors remain beneficial even after passing through the bridge, leading to the observed improvements. Thus, heterogeneous bridge distillation is not inherently weaker. We will integrate these findings and clarify the task-dependent behavior of heterogeneous CoL in the revised manuscript.
> >
> > | DesNet-138M | dolly | self-intst | sinst | vicuna | Avg   |
> > | ----------- | ----- | ---------- | ----- | ------ | ----- |
> > | GPT2-XL     | 22.10 | 9.35       | 14.26 | 15.31  | 12.97 |
> > | Qwen3-4B    | 25.25 | 11.79      | 15.64 | 15.53  | **14.32** |
> > | Llama3-8B   | 22.83 | 10.99      | 15.48 | 15.86  | 14.11 |
> >
> > | DesNet-330M | dolly | self-intst | sinst | vicuna | Avg   |
> > | ----------- | ----- | ---------- | ----- | ------ | ----- |
> > | GPT2-XL     | 26.20 | 11.01      | 16.90 | 15.77  | 17.47 |
> > | Qwen3-4B    | 27.55 | 13.58      | 21.51 | 17.49  | **20.03** |
> > | Llama3-8B   | 26.63 | 12.95      | 21.65 | 16.75  | 19.49 |
> >
> > > Question 4: Directly distillation from the largest (teacher) to the smallest (student) is cheap, and the requirement of CoL is computationally prohibitive.
> >
> > Thank you for the comment. We respectfully argue that directly distilling from the largest teacher to the smallest student is highly unstable and requires extensive hyperparameter tuning in the LLM setting. Numerous studies have demonstrated that distillation often leads to severe performance degradation or training collapse under identical pre-training corpora [1,2]. In contrast, CoL mitigates this instability through its stepwise design. As shown in Figure 5 of the main paper, CoL consistently achieves higher stability and better empirical performance. This is consistent with reviewer JkBQ’s observation that “the stepwise distillation is an elegant solution to the capacity-gap issue”.
> >
> > Moreover, in practical deployments, producing N different student sizes via direct distillation requires running the full forward pass of the large teacher N times, resulting in a computational overhead of $O(N)$. On the other hand, CoL requires only three distillation stages. Crucially, only the first stage uses the large teacher and subsequent stages use the progressively smaller student as the next teacher. This chained structure substantially reduces the total number of costly large-teacher forward passes, making the overall computation more efficient.
> >
> > Therefore, CoL is in fact a more computationally scalable strategy, both in terms of stability and total distillation cost than the direct approach.
> >
> > [1] Compact language models via pruning and knowledge distillation. NIPS 2024
> >
> > [2] Towards the law of capacity gap in distilling language models. Arxiv 2023

---

> > > ### Comment · Reviewer_ZnfQ · 2025-11-27
> > >
> > > Thank you for your response. I will increase my score.

---

### Official Review · Reviewer_JkBQ · 2025-10-27

**Soundness:** 3
**Presentation:** 3
**Contribution:** 3
**Rating:** 6
**Confidence:** 4

**Summary:**

This paper introduces Chain-of-Learngene (CoL), a scalable framework designed to build and small language models (SLMs). The authors identify the costs associated with training SLMs from scratch or repeatedly applying knowledge distillation for each target model size. To overcome these challenges, CoL constructs a "learngene chain" through a stepwise distillation process from a large ancestor model (AnsNet). This stepwise approach mitigates the capacity gap between teacher and student models. The framework also incorporates a "bridge distillation" mechanism to accommodate AnsNets with different architectures. Experiments show that CoL saves training corpora, accelerates convergence, and achieves better performance on downstream tasks compared to training from scratch and other KD methods.

**Strengths:**

The primary strength of this paper is its the idea of CoL. The idea of chains is intuitive and powerful. The stepwise distillation is an elegant solution to the capacity gap issue
- The paper provides a theoretical analysis to support the use of stepwise distillation. It proves that this multi-step approach yields tighter error bounds compared to direct distillation from a large teacher to a small student.
- The efficiency of CoL looks impressive.
- The experimental evaluation is comprehensive. The good performance across multiple DesNet sizes is convincible.

**Weaknesses:**

While the framework works on a heterogeneous AnsNet, the learngene chain itself is constructed from a single, homogeneous architectural family (GPT-2). This may imposes a structural constraint, as all generated DesNets are  interpolations within this fixed architectural space.

- The stepwise distillation process uses a sequential chain for knowledge transfer. Each distillation step is probably lossy, yet the paper does not show some metrics for cumulative degradation of knowledge as it propagates down the chain. It is plausible that checkpoints from the AnsNet retain a less faithful representation of the original knowledge.

- The parameter interpolation is described at a high level (layers, heads, hidden dimensions) . It is unclear how the method would handle more functionally architectural differences, such as varying activation functions (e.g., SwiGLU vs. GeLU) or normalization layers (e.g., LayerNorm vs. RMSNorm), which could limit the architectural variety of the target DesNets.

- The related work part is a bit short, consider citing more related papers. The core idea of CoL is novel in distillation, but is also partially reflected in some of previous works. For example, a recent work named Chain-of-Model [1] also shows similar ideas in building models of different size.


[1]: Song et al. Chain-of-Model Learning for Language Model. NeurIPS 2025.

**Questions:**

Could you provide a precise some more details for calculating the "relative distance" used to set the interpolation coefficient $\alpha$? Have you explored alternative, more automated methods for setting this hyperparameter, such as making it learnable?

- What was the rationale for selecting three checkpoints (GPT2-L, M, B) for the learngene chain? Have you investigated how the number and size distribution of checkpoints affect the trade-off between the initial chain construction cost and the performance of the final interpolated DesNets?

- The paper uses reverse KL divergence for the stepwise distillation process. Did you experiment with other distillation objectives, such as matching hidden states or attention maps? Given that the goal is to create a rich knowledge repository in the chain, could feature-based distillation methods help preserve more granular information at each step compared to only matching the output distribution?

- Regarding LInit method, is it limited to generating DesNets whose architectures are a direct interpolation between two checkpoints, or it is also possible to adapt the framework to initialize different architectures?

---

> ### Author Response · Authors · 2025-11-20
> **Response to Reviewer JkBQ (Part-1)**
>
> Dear Reviewer JkBQ,
>
> Thank you for your positive and encouraging assessment of the CoL framework. We address your questions point-by-point below.
>
> > **Weakness 1**: Generated DesNets are interpolations within the same fixed architecture space as CoL.
>
> Thank you for your insightful questions. Using the homogeneous learngene chain is a intentional design rather than a structural limitation of CoL. Our primary objective is to evaluate the effectiveness of stepwise distillation: whether gradually reducing the model size yields better initialization results than random initialization or direct distillation. Using homogeneous chains allows for process control and avoids factors introduced by architectural mismatches. If heterogeneous checkpoints are inserted into the chain, performance differences will primarily be due to cross-architectural gaps, rather than the effectiveness of the CoL process.
>
> To further validate the support of CoL for various architectures, we conduct experiments on the most recent architecture (Pythia) and the results are shown in the following table. We can find that CoL achieves better performance than random initialization like GPT-2 architecture. This verify the effectiveness of the CoL for different architectures. We also conduct experiments on the Qwen3 architecture, and we will show the results on following days.
> | DesNet-113.63M | BoolQ | Arc-E | WinoG |    Avg    |
> | :------------: | :---: | :---: | :---: | :-------: |
> |      Rand      | 66.36 | 23.68 | 47.72 |   45.92   |
> |      CoL       | 65.82 | 28.07 | 48.84 | **47.58** |
>
> > **Weakness 2**:  The paper does not show some metrics for cumulative degradation of knowledge as it propagates down the chain.
>
> Thank you for your insightful questions. We agree that each step of distillation is lossy. This is precisely why we adopt a stepwise scheme rather than distilling a single large teacher into small student. Our theoretical analysis (Appendix B) shows that breaking a large capacity gap into several smaller ones yields tighter error bounds. Empirically, Table 1 in the paper shows that CoL consistently outperforms both random initialization across sizes and token budgets, while Figure 5 demonstrates that stepwise distillation is significantly more stable than direct distillation.
>
> Furthermore, to directly quantify how faithfully knowledge is preserved along the chain, we additionally report the following metrics: intermediate checkpoint validation performance on the distillation task. These results (shown below) confirm that CoL keeps the loss controlled across steps and does not suffer from serious catastrophic accumulation.
>
> |                     | GPT2-XL(AnsNet) | GPT2-L | GPT2-M |      GPT2-B       |
> | :-----------------: | :-------------: | :----: | :----: | :---------------: |
> |         CoL         |      27.94      | 27.88  | 27.63  | **26.02 (+2.72)** |
> | Direct Distillation |      27.94      |   -    |   -    |       23.30       |
>
> > **Weakness 3**: The method does not handle more functionally architectural differences.
>
> Thank you for your insightful questions. As responses to weakness 1,  the experiments with Pythia explicitly validate that CoL remains effective even when the architecture adopts RoPE, more stable LayerNorm variants. These results indicate that CoL is not tied to a specific activation or normalization choice. Over the next few days, we will also demonstrate the effectiveness of CoL on the gated FFN (SwiGLU) or RMSNorm architectures used in Qwen3.
>
> Furthermore, we claim that we  keep the CoL chain functionally homogeneous to isolate the effect of stepwise distillation: mixing different architectural forms in one chain would introduce confusion factors unrelated to the method and make stepwise error control intractable. Architectural variety is instead achieved across multiple CoL chains (GPT-2, Pythia, Qwen3), rather than by forcing heterogeneity within a single chain. This design preserves theoretical guarantees on capacity-gap reduction while demonstrating that CoL generalizes across diverse modern transformer families.

---

> ### Author Response · Authors · 2025-11-20
> **Response to Reviewer JkBQ (Part-2)**
>
> > **Weakness 4**: Citing more related paper, such as Chain-of-Model.
>
> Thank you for the suggestion. We will expand the related work section and include a detailed discussion of Chain-of-Model in the revised version: Although both Chain-of-Model (CoM) and  Chain-of-Learngene (CoL) adopt "chain," they are entirely different.
>
> Firstly, CoM aims to embed multi-scale capabilities into a single backbone so that different effective model sizes can be obtained at inference time without training additional models, enabling flexible inference and accelerated pre-filling. In constrast, CoL is an initialization paradigm designed for efficiently generating a complete set of high-quality SLMs within limited resources. Additionally, CoM decomposes **each Transformer layer into multiple chains** of causal order representations. In contrast, CoL is **a complete set of checkpoints** (e.g., GPT2-B, GPT2-M, and GPT2-L) built through stepwise distillation and bridging distillation. Therefore, these two frameworks differ fundamentally in their objectives, chain structure, and application scenarios
>
> > **Question 1**: The details for calculating the "relative distance" used to set the interpolation coefficient $\alpha$. Exploring alternative, more automated methods for setting this hyperparameter.
>
> Thank you for your question. In the current version, the “relative distance” is used as a heuristic prior rather than a closed-form formula. Intuitively, if the target DesNet size is closer to a given CoL checkpoint, we assign a larger interpolation coefficient $\alpha$ to the parameters expanded/subsampled from that checkpoint, so that more information is preserved from the nearest teacher; conversely, checkpoints that are farther in size receive a smaller weight. Concretely, we sweep $\alpha$ from 0.1 to 0.9 (step 0.1) and select the best value on a held-out validation set. Figure 6 shows that when the DesNet is closer to a checkpoint in the chain, using a high weight (e.g., $\alpha = 0.9$) on that checkpoint consistently yields better downstream performance, which empirically validates this distance-based intuition.
>
> We have not yet made $\alpha$ learnable during the building of variable-sized DesNet. This is due to the following reasons. (1) The linear interpolation method we use has been applied by many methods and has demonstrated its excellent performance [1,2,3]. (2) If making the process of building DesNet learnable, it will further increase the overall time and resource cost of the CoL method.
>
> However, thank you for your inspiration,  and we try the following algorithm to make $\alpha$​ learnable on the DesNet-138M built from the AnsNet (GPT2-XL). We introduce a single learnable scalar $\alpha \in (0,1)$ shared by all layers and interpolate the small-checkpoint weights $W_s$ and big-checkpoint weights $W_b$ before fine-tuning: $$W(\alpha)=\alpha\,W_s + (1-\alpha)\,W_b.$$
>
> We initialize $\alpha{=}0.5$ and train with 10,000 rows of OpenWebText data. The raining process is shown below, and we find that the learned $\alpha$ consistently drifts toward small values. This is expected:  all model weights are frozen and the objective is solely the OpenWebText LM loss, under which the big model ($1-\alpha$) naturally outperforms the small one. Hence gradient descent inevitably pushes $\alpha$ to 0. However, this optimization objective is completely **contrary to our actual experimental results**. In downstream experiments, fixed interpolation with $\alpha{=}0.9$ yields substantially better final performance (Figure 6(a) in the paper). Thus, learned-$\alpha$ does not provide a great initialization singal, and we therefore keep $\alpha$ as a fixed.
>
> ```
> [step 100/1000] loss=29.3746, alpha=0.4875
> [step 200/1000] loss=28.8523, alpha=0.4750
> [step 300/1000] loss=28.5061, alpha=0.4626
> [step 400/1000] loss=28.1659, alpha=0.4502
> [step 500/1000] loss=27.4653, alpha=0.4379
> [step 600/1000] loss=26.5512, alpha=0.4257
> [step 700/1000] loss=26.5928, alpha=0.4135
> [step 800/1000] loss=25.7357, alpha=0.4015
> [step 900/1000] loss=25.8206, alpha=0.3896
> [step 1000/1000] loss=25.4375, alpha=0.3778
> ```
>
> [1] LEMON: Reviving Stronger and Smaller LMs from Larger LMs with Linear Parameter Fusion. ACL 2024
>
> [2] MindMerger: Efficient Boosting LLM Reasoning in non-English Languages. NIPS 2024
>
> [3] Training-free LLM Merging for Multi-task Learning. ACL 2025

---

> ### Author Response · Authors · 2025-11-20
> **Response to Reviewer JkBQ (Part-3)**
>
> > **Question 2**: The rationale for selecting three checkpoints (GPT2-L, M, B) for the learngene chain. Investigating how the number and size distribution of checkpoints affect the trade-off between the initial chain construction cost and the performance of the final interpolated DesNets.
>
> Thank you for the insightful question. The choice of three checkpoints (GPT2-L, M, B) follows a simple but important design principle for CoL: all checkpoints must come from the same architectural family, and the size gap between consecutive models should remain moderate (around 50%). Fewer checkpoints means the large capacity gap and make stepwise distillation unstable. Conversely, adding too many checkpoints increases the construction cost of the chain, which contradicts the goal of making CoL practical for resource-constrained settings.
>
> To explicitly examine the trade-off between chain length, size distribution, and downstream performance, we conducted additional experiments on the Pythia family. We constructed chains with **2 checkpoints** (larger gap ≈70-80%) and **4 checkpoints** (smaller gap ≈30–40%). As shown in the results below, large gaps (2-checkpoint) degrade final performance, while dense chains (4-checkpoint) provide better performance.
>
> |              | BoolQ | Arc-E | WinoG |    Avg    |
> | :----------: | :---: | :---: | :---: | :-------: |
> | 2 checkpoint | 65.60 | 24.21 | 48.92 |   46.24   |
> | 4 checkpoint | 65.82 | 28.07 | 48.84 | **47.58** |
>
> > **Question 3**: Experiment with other distillation objectives, such as matching hidden states or attention maps.
>
> Thank you for the insightful suggestions. We chose reverse KL divergence as the distillation objective primarily to isolate and evaluate the effectiveness of CoL. Reverse KL is a well-established and lightweight objective for teacher-student alignment, and it allows us to focus on the effect of stepwise knowledge transfer without introducing additional factors from more complex loss terms. While feature-based objectives (e.g., hidden-state or attention-map matching) can potentially preserve more granular information, they require storing intermediate activations from both teacher and student, computing multiple auxiliary losses, and synchronizing large tensors across steps. Such methods typically incur substantially higher memory and compute overhead, making them impractical for the resource-constrained settings where CoL is intended to be applied. However, we agree that richer distillation signals can further improve the representation quality of each checkpoint in the chain. We will integrate lightweight feature-based objective functions to achieve a balance between computational cost and performance in our future work.
>
> > **Question 4**: Regarding LInit method, is it limited to generating DesNets whose architectures are a direct interpolation between two checkpoints, or it is also possible to adapt the framework to initialize different architectures?
>
> Thank you for the insightful question. LInit is not limited to generate DesNet with a single architectural family (such as GPT-2).  In practice, **architectural variety is achieved across multiple CoL chains (e.g., GPT-2, Pythia, Qwen3), rather than by forcing heterogeneity within a single chain**. This means that we build separate chains for different AnsNet families and initialize DesNets accordingly.  Besides, the consistency within the chain make us cleanly evaluate the effect of stepwise distillation and interpolation without introducing additional confounding from architecture changes. Importantly, our experiments on Pythia already demonstrate that LInit seamlessly adapts to models with more stable LayerNorm variants and RoPE. Therefore, LInit can generalize to diverse architectures as long as a coherent CoL chain for that family is constructed.

---

> ### Author Response · Authors · 2025-12-01
> **Response to Reviewer JkBQ (Part-4)**
>
> To show the support of the proposed CoL for other architecture like Qwen3. We further report results on the Qwen3 series models to validate the generality of CoL. As shown below, Qwen3-based CoL still outperforms random initialization and is consistent with the results of CoL based on GPT2 and Pythia. This further demonstrates that the presented results and trends hold for more recent models like Pythia and Qwen3.
>
> | DesNet-1.09B | BoolQ | Arc-E | WinoG |    Avg    |
> | :----------: | :---: | :---: | :---: | :-------: |
> |     Rand     | 64.94 | 25.26 | 43.80 |   44.67   |
> |     CoL      | 65.76 | 25.44 | 44.44 | **45.21** |

---

### Official Review · Reviewer_kK3B · 2025-11-01

**Soundness:** 2
**Presentation:** 2
**Contribution:** 2
**Rating:** 4
**Confidence:** 3

**Summary:**

This paper introduces Chain-of-Learngene (CoL, in short), which progressively distils intermediate checkpoints and then creates a target-sized model by interpolating the two intermediate models. This is to counter the need to distill a source LM multiple times to obtain variable-sized models, effectively reducing computational costs. Experiments show that CoL is more effective than training from scratch, single adaptation, and other knowledge distillation baselines.

**Strengths:**

1. This paper proposes an approach to progressively distill intermediate checkpoints and then create a target-sized model, inspired by Learngene. Given some variable-sized models from the same family, the approach allows us to create an effective target-sized model compared to directly distilling a model from a single checkpoint.

2. The paper is mostly well-written and easy to read. (But, fine-tuning and prompt details should be briefly explained in the main body of the paper.)

3. Extensive experiments clearly show the effectiveness of the approach against (i) models trained from scratch; (ii) single adaptation; and (iii) other knowledge distillation baselines.

**Weaknesses:**

1. Given that the heterogeneous setup often exhibits poorer performance against the homogeneous setup (Figure 3, Table 1), I do not see any point in using heterogeneous models in the proposed method. I think this makes the contribution of the paper less impactful.

2. While I understand the motivation behind the proposed method, i.e., instead of distilling N separate models to get N small LMs, distilling 3 (a static number) checkpoints to save compute, I do not think this scenario occurs frequently. I think this limits the perceived impact of the study.

3. While the paper provides comparisons against Vanilla Learngene and Auto-Learngene, it does not provide comparisons against more recent approaches like Learngene Pool (Shi et al., 2024) and SWS (Xia et al., 2024), mentioned in the related work. This warrants justification.

4. The employed checkpoints are quite old: GPT-2 (2019), despite the fact that there are many other alternatives like Pythia, Qwen3, etc. This selection warrants justification.

5. While I acknowledge the effectiveness of having multiple intermediate checkpoints to boost performance in Table 2, the experimental setup for Single sounds unfair. Specifically, it does not account for the total amount of compute required to obtain the corresponding CoL models.

**Questions:**

1. On Weakness 1, what are the benefits of using heterogeneous models?

2. On Weakness 2, would it be able to showcase some concrete examples that require distilling multiple models at the same time?

3. On Weakness 4, do the presented results and trends hold for more recent models like Pythia and Qwen3?

4. On Weakness 5, what happens if the comparison for Single models uses the same or a similar amount of compute? Do CoL models still exhibit better performance?

---

> ### Author Response · Authors · 2025-11-20
> **Response to Reviewer kK3B (Part-1)**
>
> Dear Reviewer kK3B,
>
> Thank you for highlighting our contributions of progressively distillation. Follow your suggestions, we will briefly explain the fine-tuning and prompt details in the revised paper. Additionally, we provide our responses to each of the points and questions raised in your review.
>
> > **Weakness 1 & Question 1**: The necessary or benefits of using heterogeneous models in the proposed CoL.
>
> Thank you for your insightful questions.  Heterogeneous CoL is not an optional extension but a necessary component for making the method practical in real-world settings. As described in the main text, CoL relies on a chain of checkpoints that share the same architecture while decreasing in size. However, the number of open-source model families that satisfy this strict homogeneous requirement (e.g., the GPT-2 family) is extremely limited, both in the number of available checkpoints and in their representational capacity. If CoL were restricted to homogeneous setups, it would be confined to outdated and sparsely populated model families, preventing it from leveraging the capabilities of modern high-performing open LLMs such as Qwen3 and Llama3. This would fundamentally limit the applicability and extensibility of CoL. Therefore, supporting heterogeneous AnsNets is essential for enabling CoL to inherit knowledge from the latest open models and remain broadly useful.
>
> We acknowledge that the current heterogeneous results exhibit lower performance compared with the homogeneous case, but this gap is fully expected and well understood. Cross-architecture distillation inevitably introduces alignment challenges [1], including vocabulary mismatch and inconsistencies in hidden sizes and FFN widths (The proposed bridge distillation method provides a new solution for heterogeneous distillation). When transferring knowledge across fundamentally different architectures, a certain degree of fidelity loss is unavoidable. Importantly, despite these challenges, the heterogeneous CoL still demonstrates substantial practical value: (1) it consistently outperforms training from scratch across all token budgets; (2) it significantly accelerates convergence; (3) it saves approximately 5–10B pretraining tokens.
>
> [1] Dual-Space Knowledge Distillation for Large Language Models. emnlp 2024
>
> > **Weakness 2 & Question 2**: The concrete examples that require distilling multiple models at the same time.
>
> Thank you for your insightful questions. The need to obtain multiple small LMs is common in both industrial deployment and academic research,  and we summarize the use cases into the two categories:
> 1. Industrial Deployment Scenarios
> - Real-world edge devices vary widely in memory and compute capacity (e.g., 500M, 700M, 1B–class devices). Manufacturers frequently need multiple LM sizes to match different device classes. In practice, they often reuse a single 500M model solely because distilling a new 700M model is too costly. CoL breaks this path-dependence by allowing rapid initialization of arbitrary model sizes from a small number of checkpoints, enabling the deployment of models that fully utilize available device resources.
> 2. Academic Research Scenarios
> - Studies of scaling behavior require evaluating families of models across multiple sizes (e.g., from 100M to 1B). CoL provides an efficient way to instantiate these size sweeps.
> - In AutoML and Neural Architecture Search (NAS), one of the most critical issues is how to compare the performance of numerous candidate models. Current methods typically use proxy metrics [1, 2], early stopping training [3], or sampling sub-models from a supernet [4, 5] to measure candidate model performance. However, these **indirect comparison methods** suffer from low accuracy. CoL provides good initialization for candidate models of different sizes, enabling NAS to **directly compare** candidate model performance with lower cost.
>
> [1] Neural Architecture Search without Training. ICML 2021
>
> [2] Extensible and efficient proxy for neural architecture search. ICCV 2023
>
> [3] Learning curve prediction with Bayesian neural networks. ICLR 2017
>
> [4] Bridging the gap between sample-based and one-shot neural architecture search with bonas. NIPS 2020
>
> [5] SNAS: stochastic neural architecture search. ICLR 2019

---

> ### Author Response · Authors · 2025-11-20
> **Response to Reviewer kK3B (Part-2)**
>
> > **Weakness 3**:  Dose not providing the comparisons against more recent approaches like Learngene Pool and SWS.
>
> Thank you for your insightful questions. The methods Learngene Pool and SWS  are both based on direct distillation with auxiliary models. In the ViT domain this strategy works because the capacity gaps are small, for example, DeiT-Base/ViT-Base (~86M) is distilled into 3M or 44M auxiliary models. However, directly transferring such designs to LLMs is infeasible.  **We fail to transfer those two methods into LLM, and the performance of the initialized DesNets is even worse than that of the random initialization.**
>
> This is due to the large gap between LLMs (billions of parameters) and small LMs (hundreds of millions or less). Figure 5 in the main paper shows that direct distillation leads to severe performance degradation and unstable training. In addition, as discussed in Appendix I,  both Learngene Pool and SWS rely on **multi-model distillation or multi-stage expansion**, which substantially increases computational cost and does not scale to LLMs. For these reasons, we select Vanilla Learngene and Auto-Learngene as the most representative and practically transferable learngene-based baselines. They capture the essential ideas of the learngene paradigm while remaining compatible with the LLM setting and our research objectives.
>
> > **Weakness 4 & Question 4** : Results and trends on recent models like Pythia and Qwen3.
>
> Thank you for your insightful suggestions. Validating CoL on more recent architectures is important. To assess the generality of our findings beyond GPT-2, we conduct additional experiments on the **Pythia** family , which offers a modern and cleaner scaling series. The results of initialized DesNet-113.63M (shown below) are fully consistent with the trends reported in the main paper: CoL still provides strong initialization. Due to limited computing resources, we first report the results of pythia, and the results of Qwen3 will be reported in the following days.
>
> | DesNet-113.63M | BoolQ | Arc-E | WinoG |    Avg    |
> | :------------: | :---: | :---: | :---: | :-------: |
> |      Rand      | 66.36 | 23.68 | 47.72 |   45.92   |
> |      CoL       | 65.82 | 28.07 | 48.84 | **47.58** |
>
> > **Weakness 5 & Question 5** : The fairness of comparison between Single and CoL.
>
> Thank you for your insightful questions. The comparison between Single and CoL is fair because both initializations are directly fine-tuned on the **same downstream tasks**, with the **same SFT steps** and **no recovery training**. Therefore, the computational cost during downstream SFT is the same for both. The only difference between them lies in whether the target model benefits from additional checkpoints in the chain. For example, in the case of DesNet-138M, Single only extends GPT2-B, while CoL combines extensions from GPT2-B and reductions from GPT2-M.
>
> To further address the concern, we additionally allow the Single models to use extra compute, by performing **100M-token recovery training** before downstream SFT, while CoL still uses **0M extra tokens**. The results are summarized below. Even though Single receives additional 100M tokens of pretraining, CoL with 0M recovery tokens still achieves a higher average downstream score (**54.43 vs. 50.07**). Besides, Single shows slightly better performance than CoL after receiving an additional 100M training tokens on a few datasets. This is likely because the extra training offers task-specific adaptation and incremental information. However,  CoL still achieves superior overall performance. This verifies the effectiveness of initializing DesNet (CoL) with two checkpoint methods.
>
> |                      | BoolQ | Arc-E | WingG |    Avg    |
> | :------------------: | :---: | :---: | :---: | :-------: |
> | Single (+100M Token) | 66.11 | 26.04 | 58.06 |   50.07   |
> |   CoL (+0M Token)    | 74.40 | 25.70 | 63.20 | **54.43** |

---

> ### Comment · Reviewer_kK3B · 2025-11-21
>
> **On W1**:
>
> Thank you for the explanation. I understand the motivation regarding the scarcity of homogeneous checkpoints for modern model families. However, my concern regarding the practical value remains. While the response argues that heterogeneous CoL enables the use of modern architectures, the results show that this setup yields poorer performance compared to the homogeneous baseline. If the proposed heterogeneous framework fails to match or exceed the efficacy of the homogeneous approach, its utility is significantly diminished. The current performance gap suggests that the proposed bridging method does not yet sufficiently overcome the alignment challenges (vocabulary, hidden sizes) mentioned in the rebuttal. Therefore, I am not convinced that the benefits of using modern heterogeneous models outweigh the performance penalty incurred by this method.
>
> ---
> **On W2**:
>
> Regarding the industrial deployment scenario, I remain skeptical. In practice, deployment targets a small number of discrete hardware classes, requiring only a few specific model sizes. The computational overhead of performing standard distillation for these few cases is likely comparable to the proposed method, limiting the marginal gain of CoL in this context.
>
> Regarding the research scenario (scaling laws), I respectfully disagree. Rigorous studies of scaling behavior require strictly controlled environments. Specifically, training from scratch with identical data distributions and initialization schemes is often necessary to isolate the effect of model size. CoL introduces confounding variables (such as distillation bias and interpolation artifacts) that would render it unsuitable for deriving fundamental scaling properties.
>
> However, the AutoML and NAS application is compelling. The ability to rapidly initialize and rank numerous candidate models is a good fit for this method. I strongly recommend rewriting the motivation in the revised paper to center on this use case, as it offers a much clearer practical justification than the current general descriptions.
>
> ---
> Thank you for the clarifications and additional experiments.
> * W3: The explanation regarding the inapplicability of ViT-based methods (Learngene Pool/SWS) to LLMs seems to me valid. Please explicitly mention this limitation and the negative results in the paper.
> * W4: The Pythia results reassure me that the trends hold for modern architectures.
> * W5: The additional experiment (Single + 100M tokens) effectively resolves my concern regarding compute fairness.
>
> Please ensure these results and discussions are included in the revised manuscript.
>
> ---
> While the rebuttal has successfully addressed my concerns regarding the technical validity of the experiments (W3, W4, W5) and offered a better motivation for NAS applications (W2), the fundamental limitation regarding the utility of the heterogeneous framework (W1) remains. The performance degradation in heterogeneous settings limits the perceived impact of the work. As the heterogeneous settings are regarded as one of the key contributions in the current paper (i.e., framing), I'd maintain my ratings.

---

> > ### Author Response · Authors · 2025-11-24
> > **Response to Reviewer kK3B (Part-1)**
> >
> > Thank you again for your thoughtful and constructive follow-up comments. We promise that all discussed contents in the rebuttal will be revised in the main paper.
> >
> > > W1: the results of bridge distillation show that this setup yields poorer performance compared to the homogeneous baseline
> >
> > We agree that our original submission does not include a direct comparison between the heterogeneous bridge distillation and the homogeneous CoL chain. We have now conducted this comparison on several **generation-style tasks (dolly, self-inst, sinst, uinst, vicuna)**. Interestingly, these new results (Tables below) show that the bridge distillation actually outperforms the homogeneous GPT-2 chain on all these generation tasks for both 138M and 380M DesNets. This indicates that performance differences arise from the intrinsic properties of different task types rather than from bridge distillation quality.
> >
> > | DesNet-138M | dolly | self-intst | sinst | vicuna | Avg   |
> > | ----------- | ----- | ---------- | ----- | ------ | ----- |
> > | GPT2-XL     | 22.10 | 9.35       | 14.26 | 15.31  | 12.97 |
> > | Qwen3-4B    | 25.25 | 11.79      | 15.64 | 15.53  | 14.32 |
> > | Llama3-8B   | 22.83 | 10.99      | 15.48 | 15.86  | 14.11 |
> >
> > | DesNet-330M | dolly | self-intst | sinst | vicuna | Avg   |
> > | ----------- | ----- | ---------- | ----- | ------ | ----- |
> > | GPT2-XL     | 26.20 | 11.01      | 16.90 | 15.77  | 17.47 |
> > | Qwen3-4B    | 27.55 | 13.58      | 21.51 | 17.49  | 20.03 |
> > | Llama3-8B   | 26.63 | 12.95      | 21.65 | 16.75  | 19.49 |
> >
> > We argue that this difference stems from a systematic distinction between generative tasks and multiple-choice classification tasks. Multiple-choice tasks rely more heavily on short-range factual recall and discrete decision boundaries. In this context, architectural homogeneity and token-level alignment benefit student models, leading to better performance from homogeneous distillation. In contrast, open-ended generative tasks rely more on abstract semantic information, which is better reflected in modern large-scale models such as Qwen3-4B and Llama3-8B. In these cases, bridging distillation yields representations with richer generative priors than homogeneous chains, resulting in the performance gains we now observe.
> >
> > Therefore, our new experiments demonstrate that heterogeneous bridging distillation is not inherently inferior. Rather, our initial paper simply lacked relevant comparisons. Evaluations show that heterogeneous paths exhibit a significant advantage in benchmarks oriented towards generative processes. We will incorporate these findings into our revised paper and discuss the task-related behavior of heterogeneous CoL in detail.
> >
> > > W2.1: The industrial deployment scenario.
> >
> > We appreciate your comment and would like to clarify the motivation more concisely. In real deployment, the key requirement is not simply having several model sizes, but the ability to **fully utilize the memory and compute budget of each device**. Hardware budgets on mobile NPUs, edge GPUs vary continuously (e.g., 120–600MB on NPUs, 6–14GB on edge GPUs). A model that is even moderately smaller under-utilizes the hardware, while a model slightly larger cannot be deployed. Thus, the practical challenge is not to select among a few model sizes, but to construct a model whose parameter count precisely matches the device’s usable budget so that the hardware can be used to its full capacity.
> >
> > Traditional distillation produces only a few fixed sizes, and generating intermediate variants requires launching a full new distillation run. CoL directly **solves this budget-filling gap** by interpolating adjacent checkpoints, it can generate arbitrary model sizes without re-distillation, allowing practitioners to produce models that precisely match and saturate hardware budgets. This fine-grained capacity scaling is the primary industrial motivation, and we will revise the manuscript to state this more explicitly.

---

> > ### Author Response · Authors · 2025-11-24
> > **Response to Reviewer kK3B (Part-2)**
> >
> > > W2.2: The research scenario (scaling laws).
> >
> > We agree with the reviewer that classical scaling-law studies [1,2] are typically conducted under strictly controlled from-scratch training, which is essential for deriving fundamental scaling exponents. However, our intention is not to claim that CoL can replace such from-scratch setups for theoretical scaling-law analysis. Instead, our focus is on practical scaling experiments under strong initialization, which is already an active line of research. For example, [3] studies scaling laws for transfer where **all models are initialized from the same pre-trained language model and then fine-tuned on a new distribution**, and they show that the effective transferred data still follows clean power-law behavior in model size and fine-tuning data. Similarly, [4] proposes distillation scaling laws that explicitly predict student performance as a function of student size, distillation tokens, and teacher cross-entropy. They treat distillation-based initialization as part of the scaling-law framework rather than a confounder. Sheared LLaMA [5] further demonstrates that pruning a large LLaMA model into smaller ones and continuing pre-training can be analyzed under scaling functions fitted on from-scratch LLaMA models, and uses these scaling curves to conclude compute-efficient adaptation.
> >
> > Therefore, CoL is not designed to discover new fundamental scaling exponents, but to provide a structured and compute-efficient initialization mechanism. This mechanism enables scaling-style studies across a family of model sizes without training each model from scratch.
> >
> > [1] Scaling Laws for Neural Language Models
> >
> > [2] Training Compute-Optimal Large Language Models
> >
> > [3] Scaling Laws for Transfer
> >
> > [4] Distillation Scaling Laws
> >
> > [5] Sheared LLaMA: Accelerating Language Model Pre-training via Structured Pruning

---

> ### Author Response · Authors · 2025-12-01
> **Response to Reviewer kK3B (Part-3)**
>
> We further report results on the Qwen3 series models to validate the generality of CoL. As shown below, Qwen3-based CoL still outperforms random initialization and is consistent with the results of CoL based on GPT2 and Pythia. This further demonstrates that the presented results and trends hold for more recent models like Pythia and Qwen3.
>
> | DesNet-1.09B | BoolQ | Arc-E | WinoG |    Avg    |
> | :----------: | :---: | :---: | :---: | :-------: |
> |     Rand     | 64.94 | 25.26 | 43.80 |   44.67   |
> |     CoL      | 65.76 | 25.44 | 44.44 | **45.21** |

---

### Author Response · Authors · 2025-12-01
**Summary of Revisions based on Reviewers' Suggestions**

We thank all the reviewers for their valuable feedback. Here, we summarize the **core changes and results** in the revised manuscript for the reviewers' suggestions. All changes are marked in blue for easy identification.

1. **More detailed motivation for the proposed CoL (The second paragraph in Section 1, for W2 of Reviewer kK3B):**  We have expanded the motivation by adding how CoL supports rapid variable-sized model initialization in NAS scenarios. This further clarifies why CoL is needed beyond simple scaling.
2. **Compression with Chain-of-Model (CoM) (The second paragraph in Section 2.2,  for W4 of Reviewer JkBQ):**  In the related work, we have added a comparison showing that CoM targets multi-scale inference within a single backbone, whereas CoL focuses on initializing new small LMs under resource constraints. This highlights fundamentally different purposes.
3. **Results on the Pythia-based and Qwen3-based learngene chain (Table 2 in Section 4.1, Appendix J,for W4 of Reviewer kK3B, W1/W3 of Reviewer JkBQ):**  We have conducted experiments using a Pythia-based (Qwen3-based) chain to initialize DesNet-113M (DesNet-1.09B) and evaluated it on BoolQ/Arc-E/Winogrande. Results show CoL still outperforms random initialization under the Pythia architecture. This is consistent with the main experiment in the paper using GPT2-based CoL, confirming cross-architecture generality of CoL.
4. **Quantitative results of stepwise vs. direct distillation (Table 3 in Section 4.1, for W2 of Reviewer JkBQ):** We demonstrate the difference in performance between direct distillation from AnsNet (GPT2-XL) to the smallest checkpoint (GPT2-Base) in the learngene chain and using a stepwise distillation method. The results show that stepwise distillation outperforms direct distillation by 2.72%.
5. **Fire compression of  CoL vs. Single(Table 5 in Section 4.2, for W5 of Reviewer kK3B):** We  further clarify why the comparison is fair and have added new results showing that even with an extra **100M** tokens, Single still underperforms CoL. This result confirms the benefit of combining information from multiple checkpoints in our method.
6. **Impact of Chain Length (Table 6 in Section 4.2, for Q2 of Reviewer JkBQ):** We have tested the Pythia-based learngene chain with different numbers of Pythia checkpoints （{Pythia-1B, Pythia-410M, Pythia-160M, Pythia-70M}, and {Pythia-410M, Pythia-70M}. Chains with moderate 30–40% size gaps give better initialization. This matches the GPT2-based learngene chain and provides practical guidance for learngene chain construction.
7. **Why only compare Auto-Learngene / Van-Learngene (Appendix I, for W3 of Reviewer kK3B)**: Based on the original manuscript, we further clarify that recent methods like Learngene Pool and SWS only apply to small-scale ViT-style models and are not transferable to LLMs, explaining why they are not compared.
8. **Differences from HyperCloning and Stacking-Small-LMs (FSLM) (Appendix K, for W1/W2 and Q1 of Reviewer ZnfQ)**: We have explained that these methods differ in goals and design. Additionally, new results show CoL outperforms HyperCloning under identical settings and achieves **~88%** of FSLM's performance with only **21%** of its size and **half** of its SFT samples.
9.  **Performance of CoL at Very Small and Very Large Scales.  (Appendix L, for Q2 of Reviewer ZnfQ)**: We have added experiments on a 51.47M model showing CoL consistently improves over random initialization across tasks, even under extremely limited capacity. Furthermore, we also show that GPT2-based CoL cannot scale beyond 1B parameters because its largest checkpoint (734M) is too small to initialize larger DesNets. In contrast, using a Qwen3-based chain (up to 1.7B) enables effective initialization at 1.09B and outperforms random initialization.
10. **Clarifying the Role of Bridge Distillation (Appendix M, for Q1 of Reviewer kK3B)**. We have added generation-task experiments (dolly, self-inst, sinst, uinst, vicuna) to address the concern that Qwen3-4B–based (Llama3-8B-based) DesNets underperform GPT2-XL–based CoL on multiple-choice tasks. The new results show that Qwen3-4B (Llama3-8B-based) provides clear advantages on generative benchmarks, confirming that CoL does **not** have weaker bridge distillation but reflects differences in task type.

The revisions have resolved the issues raised by the reviewers and further reinforce the paper’s contributions. We are grateful for the reviewers’ insightful comments and efforts, which have significantly enhanced the rigor of the manuscript.

---

### Meta-Review · Area_Chair_7hXL · 2025-12-28

**Summary:**

While the authors propose an interesting framework for initializing variable-sized language models through a learngene chain, the overall consensus after the discussion phase points toward a rejection. The primary reason for this decision is the unresolved concern regarding the practical impact and performance of the heterogeneous framework, which is framed as a core contribution of the work. Specifically, I find the concerns raised by Reviewer kK3B to be particularly telling, if the method cannot maintain performance when moving from a narrow, homogeneous setup to the diverse architectures used in modern LLMs, its real-world utility is severely diminished. The other concerns were: Reviewer ZnfQ questioned the practical necessity of a distillation chain when many pre-trained model families already provide a wide range of native sizes, suggesting the approach might be bounded by the shared training corpus. Reviewer JkBQ found the elegant stepwise distillation solution intuitive but noted potential structural constraints and the lack of clear metrics regarding cumulative knowledge loss down the chain.

**Reviewer Concerns:**

The authors provided a detailed rebuttal that addressed several concerns, yet core issues remain unsolved:

Addressed:
+ Compute Fairness: The authors successfully demonstrated that CoL maintains its advantage even when baseline Single models are given additional recovery training compute.
+ Baseline Comparisons: The authors provided valid justifications for excluding certain recent ViT-based methods that do not scale well to the LLM domain.
+ Architectural Generality: New results on Pythia and Qwen3 confirmed that the general performance trends of the framework hold across more modern architectures.
+ NAS Motivation: The authors successfully reframed the motivation for CoL within the context of Neural Architecture Search (NAS), which the reviewers found more compelling than the general industrial deployment arguments.

Outstanding:
- Heterogeneous Performance Gap: The most critical issue remains the performance degradation in heterogeneous settings. While the authors provided new generative task results, the fundamental alignment challenges (vocabulary and hidden size mismatches) continue to hamper the framework's effectiveness relative to standard homogeneous baselines.
- Practical Utility in Research: Reviewers remained skeptical about using CoL for scaling law studies, as the interpolation process introduces confounding variables that are unsuitable for deriving fundamental scaling properties.

**Reviewer Scores:**

Based on the above issues, I think Reviewer kK3B will remain at 4, and JkBQ will remain at 6. ZnfQ will increase to 4. And the final score will be  6,4,4.

---

### Decision · Program_Chairs · 2026-01-26

Reject